# AgentAda: Skill-Adaptive Data Analytics for Tailored Insight Discovery

## Abstract

We introduce AgentAda, the first LLM-powered analytics agent that can learn and use new analytics skills to extract more specialized insights. Unlike existing methods that require users to manually decide which data analytics method to apply, AgentAda automatically identifies the skill needed from a library of analytical skills to perform the analysis. This also allows AgentAda to use skills that existing LLMs cannot perform out of the box. The library covers a range of methods, including clustering, predictive modeling, and NLP techniques like BERT, which allow AgentAda to handle complex analytics tasks based on what the user needs. AgentAda's dataset-to-insight extraction strategy consists of three key steps: a (I) question generator to generate queries relevant to user's goal and persona, a (II) hybrid Retrieval-Augmented Generation (RAG)-based skill matcher to choose the best data analytics skill from the skill library, and a (III) code generator that produces executable code based on the retrieved skill's documentation to extract key patterns. We also introduce KaggleBench, a benchmark of curated notebooks across diverse domains, to evaluate AgentAda's performance. We conducted a human evaluation demonstrating that AgentAda provides more insightful analytics than existing tools, with 48.78% of evaluators preferring its analyses, compared to 27.67% for the unskilled agent. We also propose a novel LLM-as-a-judge approach that we show is aligned with human evaluation as a way to automate insights' quality evaluation at larger scale[1].

## 1 Introduction

Large language models (LLMs) have proven to be highly effective at handling natural language tasks, but their effective integration into data analytics tasks is still a challenge. Most existing LLM-based analytics tools are general-purpose and lack the structure needed to perform advanced analytics, such as clustering, predictive modeling, or trend analysis. They often struggle with multi-step reasoning and tend to rely on basic analytical methods or requires manual intervention to select more effective techniques for a given problem. This leads to errors, inefficiencies, and an inability to handle complex workflows or domain-specific needs (de Miranda & Campelo, 2024). These limitations point to the need for more capable and structured data analytics agents that can go beyond surface-level analysis, reason through complex tasks, and adapt to the analytical demands of different tasks.

To overcome these limitations, we introduce **AgentAda**, a *skill-informed data analytics agent*. In this framework, a relevant analytical skill is retrieved from a curated skill library and used to guide the generation of executable code for the given task (see Figure 1). By equipping the LLM with well-defined, task-specific analytical methods, AgentAda moves beyond basic statistical summaries and supports more advanced forms of analysis. This helps uncover deeper, more meaningful insights, often much better than what powerful LLMs produce without access to skill information. AgentAda also adopts a structured approach to analysis by guiding the process through four key stages: question formulation, method selection, code generation, and insight extraction. Each stage is informed by the task context and aligned with the analytical goal and user persona. This structure helps the agent reason more effectively and carry out end-to-end analysis, leading to outputs that

---

[1]Codes and data are available in the supplementary materials.

are not only methodologically sound but also context-aware, actionable, and relevant to the task at hand. We observed this in our experiments, where AGENTADA consistently produced deeper and more goal-aligned insights than existing analytics agents 60.01% of times.

A major challenge in advancing LLM-based data analytics is the lack of strong evaluation frameworks that capture real-world demands. Two gaps stand out. First, current benchmarks often focus on narrow domains with simple statistical tasks (e.g., Insight-Bench (Sahu et al., 2024) focuses on business analytics) and fail to reflect the complexity of broader analytical settings. Second, there is no clear way to compare the quality of generated insights. Insight evaluation is subjective and hard to define. Human evaluation, while useful, is difficult to scale due to the expertise required. Progress needs broader, more realistic benchmarks and scalable, expert-informed evaluation methods.

To overcome the first limitation and evaluate the effectiveness of AGENTADA we introduce **KAGGLEBENCH**, a benchmark of 700 examples spanning 49 domains and 28 task types. KAGGLEBENCH addresses key limitations of prior benchmarks (Sahu et al., 2024) by covering a broader range of analytical tasks that require deeper reasoning and more advanced analytical skills. It provides a more realistic and comprehensive testbed for assessing the capabilities of LLM-based analytics agents. In addition we introduce SCORER (Structured Calibration Of Ratings via Expert Refinement), an LLM-as-a-judge framework guided by human feedback for evaluating analytical insights. Unlike prior approaches that rely on static prompts or fine-tuning, SCORER uses prompt optimization with human-annotated rubrics to achieve expert-aligned scoring, making it lightweight and scalable. To our knowledge, this is the first application of prompt-tuned LLM-as-a-judge evaluation in data

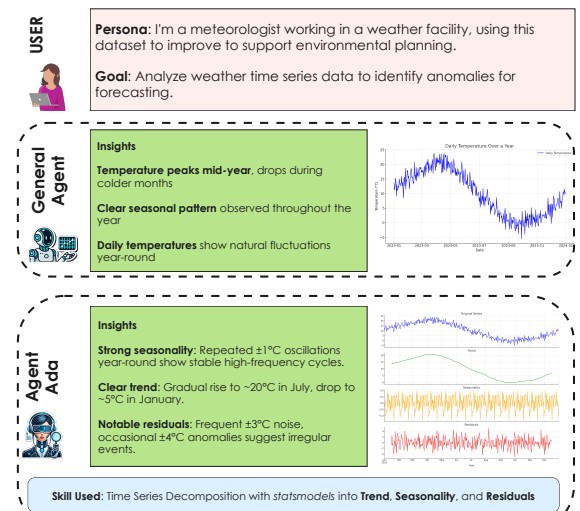

Figure 1: Unlike other data analytics agents, AGENTADA breaks down tasks into detailed, skill-specific questions aligned with the user's goal and persona, delivering deep, insightful, and factual analysis.

analytics. In our experiments, we benchmarked AGENTADA against existing agents (Hu et al., 2024; Sahu et al., 2024; Ge et al., 2023) on KAGGLEBENCH, with SCORER as the evaluation method.

Our contributions are as follows: (I) We introduce AGENTADA the first skill-informed data analytics agent equipped with a novel end-to-end pipeline that dynamically selects relevant analytical skills from a curated library and generates executable code to produce goal-aligned insights across a wide range of advanced analytical tasks. (II) We release KAGGLEBENCH, a benchmark of 700 examples spanning 49 domains and 28 task types, capturing the complexity and diversity of real-world data analysis scenarios. (III) We introduce SCORER, a novel prompt-optimized LLM-as-a-judge framework that aligns with human evaluation of analytical insights using expert-guided supervision. (IV) We conduct comprehensive evaluations showing that AGENTADA outperforms existing agents in both analytical depth and alignment with task goals and user personas.

## 2 RELATED WORK

**LLM-based Data Analytics.** Prior works on LLM-based data analytics agents have explored structured pipelines and multi-agent frameworks, but still face key limitations in adaptability, goal alignment, efficiency, and generalization. Multi-agent systems (Rasheed et al., 2024; Fischer & Biemann, 2024; Chugh et al., 2023) break down problems into sub-tasks handled by specialized agents. But they lack guidance in choosing the right analytical methods, often producing surface-level summaries or descriptive statistics rather than deeper diagnostic or prescriptive insights. They also struggle to adapt to specific user goals or personas. Other systems like InfiAgent (Hu et al., 2024) and Data Interpreter (Hong et al., 2024) use strategies like ReAct (Yao et al., 2023) and hi-

erarchical modeling to generate structured code. But without incorporating the specific analytical objectives, dataset characteristics, or examples of how domain-relevant skills should be applied, their outputs are often error prone and rely heavily on inefficient iterative debugging. In contrast, AGENTADA's skill-informed pipeline enables efficient, goal-driven, and adaptable analysis, which generalizes across various tasks and domains.

**Data Analytics Benchmarks.** Existing benchmarks for LLM-based analytics focus on narrow tasks or domains. DS-1000 (Lai et al., 2023) and DA-Code (Huang et al., 2024) target data science and agent-based tasks, while InsightBench (Sahu et al., 2024) focuses on business analytics with basic statistics. Code-centric benchmarks like LiveCodeBench and BigCodeBench (Jain et al., 2024; Zhang et al., 2024) evaluate code generation but neglect end-to-end analytics workflows. To fill this gap, we introduce KAGGLEBENCH, a multi-domain benchmark from real-world Kaggle notebooks, covering 49 domains including finance, health, and education. KAGGLEBENCH supports robust evaluation of agents like AGENTADA on complex, insight-driven analytics tasks across a wide range of domains.

**LLM-as-a-Judge Frameworks.** Most existing LLM-as-a-judge frameworks rely on static prompts or model fine-tuning, which limits their adaptability and scalability. Static prompting methods (Zheng et al., 2023; Li et al., 2023a) typically provide evaluation criteria to a powerful LLM and delegate the grading task. But, aligning with nuanced human preferences is challenging and often requires careful prompt engineering and rubric design (Zeng et al., 2023). Other approaches (Wang et al., 2023; Zhu et al., 2023; Li et al., 2023b; Kim et al., 2023) fine-tune LLMs specifically for evaluation, improving alignment with human judgment. However, these methods are often expensive and resource-intensive. More recent hybrid methods (Xu et al., 2023; Zhang et al., 2023) iteratively refine evaluators using feedback from human expert corrections. While they reduce the need for full model fine-tuning, they still involve continuous maintenance of models or example sets. In contrast to all these methods, our approach, SCORER, achieves human expert-aligned scoring purely through prompt optimization while remaining lightweight, scalable, and adaptable across analytical tasks.

## 3 KAGGLEBENCH –DATA ANALYTICS BENCHMARK

KAGGLEBENCH is a curated benchmark designed to evaluate the analytical capabilities of data analytics agents across a wide range of tasks, skills, and domains. Below, we outline the data collection and construction process in detail. See Appendix A for statistics on KAGGLEBENCH.

**Dataset Notebooks QA Generation.** The dataset is sourced from high-quality Jupyter notebooks published by data analysts on Kaggle[2] , a leading platform for data science and analysis. We collected 700 notebooks spanning diverse analytical domains and task types. Each notebook contains structured workflows, markdown summaries, and datasets, making them well-suited for insight-focused evaluation. To construct fine-grained QA examples, we parsed notebooks into cell batches and used GPT-4o to (I) generate QA pairs and (II) assign each question a task and skill label from a predefined library (Appendix B). Answers were drawn directly from markdown conclusions or code outputs, ensuring that QA pairs reflect the actual reasoning and results in the notebooks. We further verified the answer source (markdown vs. code) using a RAG-Token Model (Lewis et al., 2020). This RAG-based grounding helps maintain factuality, as QA pairs are generated directly from notebook content, making them reliable for evaluating whether models produce factually correct insights. QA pairs with invalid tasks or skills were filtered out (12.28% removed), and an LLM was used to select the top 10 diverse, well-framed QA pairs from each notebook. The prompts are provided in Appendix I.1 (OpenAI et al., 2024).

**Goal and Persona Generation.** To support goal- and persona-aware evaluation of analytical insights, we generated a corresponding *goal* and *persona* for each notebook in KAGGLEBENCH. The goal is a concise statement capturing the purpose of the analysis of the notebook, focusing on *what* and *why* something is being analyzed, without specifying *how* the analysis is performed. The persona describes the role or perspective (e.g., data analyst, business strategist) from which the analysis is conducted. An example of a generated goal and persona is shown in Figure 1. Note that while a dataset may support multiple analytical directions, we extract the goal and persona that

---

[2]https://www.kaggle.com

Figure 2: AGENTADA's pipeline for automated insights. It first generates diverse questions from the data, then uses a RAG-based skill matcher to select relevant tools. The code generator executes the analysis, answers are derived from plots and outputs, and final insights are extracted from answers which includes statistics and visualizations.

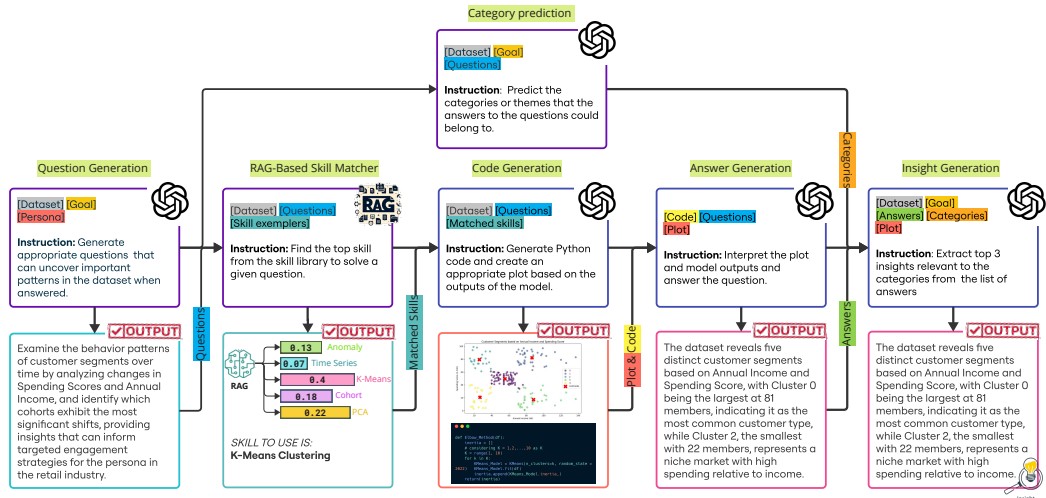

reflect the specific analysis actually carried out in the notebook. Both were extracted using GPT-4o, with the prompting strategy detailed in Appendix I.1.

# 4 AGENTADA – A SKILL-INFORMED DATA ANALYTICS AGENT

In this section, we describe the end-to-end AGENTADA pipeline (Figure 2), which consists of four stages: *Skill Matcher*, *Code Generation*, *Answer Generation*, and *Insight Extraction*. Specifically, *Skill Matcher* identifies the most relevant analytical skill for a given task, *Code Generation* produces tailored executable code, *Answer Generation* addresses each analytical question, and *Insight Extraction* summarizes and communicates meaningful results. To enable effective skill retrieval during inference, we first constructed a library of diverse analytical skills.

**Skill Set Collection.** We curated a library of 74 diverse data analytics skills, covering a broad range of tasks and algorithms 7. Implemented in Python, these skills were primarily sourced from Kaggle notebooks. To build the library, we identified high-quality notebooks across domains, converted them to markdown. Here, we retained only workflow-relevant code blocks for data preparation, modeling, evaluation, and visualization, discarding exploratory or environment-specific cells. Next, these curated code workflows were provided as input to an LLM to generate concise, text-based descriptions for each workflow. Importantly, these descriptions were not intended to merely summarize the code. They serve as the skill itself, a knowledge base that clearly explains what the algorithm does, when it should be applied, and how it can be used to solve a data analysis problem. Also, because the library is organized around modular, reusable workflows, it is naturally extensible, and new skills can be easily added as data analysis practices evolve.

Although both the skill library and KaggleBench are sourced from Kaggle, there is no data leakage between them. The skill library captures reusable algorithmic workflows and their descriptions, while KaggleBench independently constructs QA pairs from datasets. This separation ensures that the skills represent general methods, whereas the benchmark evaluates their application on unseen tasks and data.

**Dual Stage Advanced Question Generation.** Insight generation begins with asking the right questions. To guide AGENTADA in producing meaningful, goal-aligned insights for a given dataset, we start with designing a two-stage question generation process. In the first stage, we generate a set of basic data analytics questions using the dataset, goal, and persona. We focus on straightforward

tasks such as filtering or simple aggregations. In the second stage, we use the available skills in the skill library along with the generated simple questions to generate more advanced questions that require complex reasoning and advanced techniques to analyze. This setup helps AGENTADA uncover deeper patterns in the data. Some examples of both basic and advanced questions, along with corresponding analyses, are provided in Appendix I.2. Detailed prompts for both stages are provided in Appendix I.2.

**Category prediction.** To evaluate the performance of AGENTADA against other analytics agents, it is important that the insights being compared are organized around similar high-level themes. To support this, we introduce an insight category prediction module that estimates the overarching analytical themes likely to emerge from the responses to each set of questions. We achieve this by prompting GPT-4o with the dataset description, analysis goal, and the list of generated questions, asking it to predict the top three insight categories that will likely capture the essence of the responses. More details about the prompting strategy for this module are provided in Appendix I.3.

**Skill Matcher.** For each question in the advanced set, we retrieve the most relevant analytical skill from the skill library using a Hybrid Retrieval-Augmented Generation (RAG) system (Dong et al., 2024; Li et al., 2024; Su et al., 2024a; Shi et al., 2024; Sticha, 2023). This system connects natural language questions to executable analysis by combining semantic search with structured mappings between skill descriptions and their corresponding implementations. The skill matcher helps guide AGENTADA's analysis toward the most suitable techniques for answering each question accurately and efficiently. It operates in four steps: (I) an LLM interprets the question to identify its analytical intent and underlying task type, (II) the question is embedded and matched against skill summaries using OpenAI embeddings (Neelakantan et al., 2022), (III) the top-$k$ most relevant skills are retrieved from the library ($k = 3$ in our setup) and (IV) the selected skill, including its summary and implementation, is passed to the code generation module to guide the next stage of analysis. The full prompting strategy for the skill matcher is provided in Appendix I.4.

**Code Generation.** After retrieving the question and relevant skill, the code generation module produces structured, executable code to meet the analytical goal. It takes the data schema, question, predicted skill, and its summary as input to generate code with visualizations and key statistics. The skill guides the LLM in producing clear, complete code for preprocessing, analysis, plotting, and metric extraction. If execution fails, the error message is added to the prompt for regeneration, up to three attempts per question. This enables self-correction without manual input. On average, we observe $1.8$ generations per dataset with 5 questions. Full prompting details are available in Appendix I.5.

**Answer Generation.** The next step is to generate responses from the plots and statistical outputs generated by the executed code for each question generated by the question generation module. For this we use a multimodal LLM ((OpenAI et al., 2024)), that takes as input the question, generated plot, and key statistics to produce a structured response. These responses are then summarized into concise bullet points for clarity and ease of interpretation. Both answer generation prompts are provided in Appendix I.6.

**Insight Generation.** In the final insight extraction step, we aggregate the answers across all questions for a dataset and prompt the LLM with the dataset description, overall goal, generated answers, and insight categories to produce goal-aligned and actionable insights. Prompting strategies are detailed in I.7.

Figure 3: The validation loss steadily decreases during prompt optimization, indicating improved alignment between SCORER's evaluation scores and human judgments.

## 5 SCORER

We evaluate the quality of generated insights using SCORER (Structured Calibration Of Ratings via Expert Refinement), an LLM-as-a-judge framework that aligns model scores with human judgment through prompt optimization rather than fine-tuning. The key idea is that an LLM can approximate expert evaluation when guided by contextual cues from human ratings and rationales. SCORER

helps evaluate on a shared set of evaluation criteria (Section 6.1) and formulates prompt refinement as an optimization problem using TextGrad (Yuksekgonul et al., 2024). It minimizes the mean squared error between LLM and human scores. Starting from a simple starter prompt, the optimizer iteratively refines toward an expert-aligned prompt that mirrors human scoring patterns. This approach preserves the scalability of LLM-based evaluation and improves reliability and alignment with human judgment. Full prompt details are provided in Appendix J. Figure 3 shows validation loss decreasing over optimization steps, indicating closer agreement with human evaluators. More results on robustness and generalizability of SCORER is in Appendix L.

## 6 EXPERIMENTS & RESULTS

**Experimental Setup.** All LLM interactions were performed via API calls to OpenAI's GPT-4o (OpenAI et al., 2024) and text-embedding-3-small models.

### 6.1 EVALUATION OF AGENTADA'S SKILL ABILITIES

We evaluate the quality of insights generated by AGENTADA against several analytics agents, including Poirot (Sahu et al., 2024), Pandas AI (Fischer & Biemann, 2024), InfiAgent (Hu et al., 2024), MetaGPT (Ge et al., 2023), and direct prompting with GPT-4o. To test our core hypothesis that skill-informed agents produce deeper insights, we also include a variant of AGENTADA without skill guidance. In this baseline (**W/O Skill**), the LLM infers and applies skills without access to the curated skill library, while the full version is denoted **W Skill**. We compare all these agents across six rubrics: **depth of analysis**, **relevance to goal**, **persona consistency**, **coherence**, **answering question adequately**, and **plot conclusion quality** (definitions in Appendix C). When comparing the W/O Skill and W Skill variants, instead of evaluating only final dataset-level insights, we assess the quality of individual answers in the answer generation stage. This avoids the masking effect of similar-looking final outputs, which use the same LLM and prompt in both variants, and allows us to more directly capture the impact of skill guidance. Performance against other agents is reported across all rubrics, with the exception of **answering question adequately**, which was evaluated at the final insight level where questions did not align across systems.

**Human Evaluation.** In the first experiment we conducted a human evaluation of the W Skill and W/O Skill variants of AGENTADA on 100 datasets spanning diverse analytical tasks and domains. The evaluation was split into 10 batches of 10 questions, each reviewed by three independent annotators with data analytics backgrounds to ensure consistency and assess analytical depth, relevance, and reasoning quality. We used Fleiss' Kappa (Fleiss, 1971) to measure annotator agreement and assess the reliability of our evaluation (Table 8). Most criteria showed strong agreement, while *Goal Relevance* and *Persona Consistency* had lower scores, expected given their subjective nature. Annotators may avoid the "Tie" option in borderline cases, adding noise, and assessing persona alignment often depends on individual interpretation of tone and perspective. The human evaluation setup is described in detail in Appendix C. Additional statistics on the human evaluators are provided in D.

Table 2 summarizes the results of our human evaluation. Scores represent the distribution of judgments across four options: win with skill, win without skill, tie, and none. Overall, the skill-informed version of AGENTADA outperforms the W/O Skill variant across all rubrics, with the largest margin in *Depth of Analysis*, supporting our hypothesis that retrieved skills enable deeper insights. In contrast, rubrics such as Relevance to Goal and Persona Consistency show a high proportion of ties (49.22% and 61.44%), highlighting more subtle differences between the variants. This is consistent with their lower Fleiss' kappa scores, which indicate greater subjectivity.

To better understand the cases where the W/O Skill variant was judged superior, we conducted a qualitative study (Table 1). We find that when skilled responses are preferred, they typically apply the correct analytical method (e.g., PCA, collaborative filtering, market basket analysis), while unskilled answers rely on shallow heuristics such as simple counting. In the cases where unskilled answers were preferred, the skilled variant introduced unnecessary complexity. (e.g., using PageRank instead of counting connected components). Importantly, these skilled responses were not incorrect. They often applied valid analytical methods, but their complexity was not required for the question at hand, making the simpler unskilled answers more appealing. Additional qualitative analysis is provided in Appendix M.

Table 1: Qualitative comparison of AGENTADA with and without skill guidance. Skilled-preferred cases reflect correct analytical reasoning.

| Task | Question | Preferred | Unskilled Answer | Skilled Answer |
|---|---|---|---|---|
| Recommendation Systems | What trends are observed in user preferences across different categories, and how can this guide targeted recommendations? | Unskilled | Electronics and Home Appliances are the most frequently chosen categories. These can be targeted with personalized promotions. | Using Collaborative Filtering, we find latent factors influencing user-category affinity scores. The model predicts cross-domain co-preference between Electronics and Smart Gadgets. |
| Market Analysis | What purchase patterns were observed based on different days of the week? | Unskilled | Weekends show higher sales of snacks and beverages, while weekdays focus on essentials like milk and bread. | Association rule mining identifies high-confidence weekend-specific rules such as {chips} → {soda}, but this complicates a simple weekday/weekend pattern. |
| Recommendation Systems | How do the preferences of users who rate niche items overlap with general user preferences, and how can this guide recommendations? | Skilled | Users who rate niche items also seem to buy from common categories like Electronics. | Collaborative Filtering reveals that niche users share latent factors with mainstream users, allowing cross-category recommendations such as niche Book readers also preferring certain Gadgets. |
| Topic Modeling | What are the topics identified in the articles by matrix factorization? | Skilled | From word counts, I can guess the main categories are business, politics, and sports. | Latent Semantic Analysis identifies five topics with top terms: (1) business/finance, (2) tech/innovation, (3) politics/government, (4) sport/teams, (5) entertainment/media. |
| Anomaly Detection | How did adjusting the contamination parameter affect anomaly detection results? | Skilled | When we increased the parameter, we saw more anomalies flagged, but it's hard to quantify. | Increasing contamination from 0.02 to 0.06 raised detected anomalies by 48% but also increased false positives, showing a precision–recall tradeoff. |
| Market Analysis | Which product combination shows the strongest lift score? | Skilled | Customers often buy pasta with pasta sauce, so they likely have a strong relationship. | Market Basket Analysis shows Pasta + Pasta Sauce has the strongest lift of 2.8, meaning they are nearly 3x more likely to be bought together than by chance. |

Table 2: Human evaluation of insights across 100 datasets. The *Goal Relevance* rubric shows the most variation, influenced by its direct use in insight generation and the choice of analytical skills. Detailed results for the 18 evaluated tasks are provided in Tables 17 and 18 in Appendix N.

| Rubric | W Skill Win | W/O Skill Win | Tie | Neither Are Good |
|---|---|---|---|---|
| **Depth of Analysis** | **48.78** | 27.67 | 21.22 | 2.33 |
| **Relevance To Goal** | 31.33 | 17.00 | **49.22** | 2.44 |
| **Persona Consistency** | 26.11 | 10.11 | **61.44** | 2.33 |
| **Coherence** | **48.78** | 27.78 | 21.00 | 2.44 |
| **Answers Question Adequately** | **42.67** | 25.22 | 29.67 | 2.44 |
| **Plot Conclusion** | **42.00** | 23.44 | 32.33 | 2.22 |

**SCORER Evaluation.** We evaluated AGENTADA using **SCORER** on KAGGLEBENCH containing 700 datasets spanning diverse analytical tasks. To train SCORER, we first collected human evaluation scores on 100 datasets and split them into a $70/30$ train-test split. Then, the starter prompt was optimized using TextGrad (Yuksekgonul et al., 2024) to minimize the mean squared error (MSE) between the LLM-predicted scores and human evaluation scores. After optimization, the SCORER prompt achieved a validation loss of $0.4$, indicating strong alignment with human judgment and reliable replication of expert preferences. More details on SCORER human alignment is provided in Appendix H. We used the optimized human aligned prompt to score insights across all 700 datasets in KAGGLEBENCHand compare AGENTADA against other baselines. The results comparing the skill-informed (overall and top-5 frequent tasks) variant of AGENTADA and without skill variant is presented in Table 3. The most significant gains are observed in Depth of Analysis and Coherence, where over $50\%$ of the responses are rated better when guided by retrieved skills. This supports our core hypothesis. On execution-aligned rubrics like Answers Question Adequately and Plot Conclusion, the skill-informed model again performs better. This shows that guided code generation helps generate complete responses and stronger visual reasoning. Also, it is worth noting that these findings closely mirror trends observed in our human evaluation (Table 2), with high alignment across most rubrics. This further validates SCORER's effectiveness as a lightweight and scalable proxy for human judgment.

Table 3: SCORER evaluation comparing the performance of AGENTADA's W-skill (WA) variant against W/O skill (WO). Percentage results are reported across all rubrics for five representative data analytics tasks. Full results are provided in Tables 19 and 20 in Appendix P.

| Task | Depth of Analysis | | | | Relevance To Goal | | | | Persona Consistency | | | |
|------|------|------|------|------|------|------|------|------|------|------|------|------|
| | *WA* | *WO* | *T* | *N* | *WA* | *WO* | *T* | *N* | *WA* | *WO* | *T* | *N* |
| **Sentiment Analysis** | **48.65** | 27.03 | 21.62 | 2.7 | 40.54 | 13.51 | **43.24** | 2.7 | 29.73 | 10.81 | **56.76** | 2.7 |
| **Basic Data Analysis** | **51.85** | 24.44 | 20.74 | 2.96 | 33.33 | 17.04 | **47.41** | 2.22 | 31.11 | 10.37 | **55.56** | 2.96 |
| **Customer Segmentation** | **50.0** | 26.09 | 21.74 | 2.17 | 36.96 | 19.57 | **41.3** | 2.17 | 30.43 | 10.87 | **56.52** | 2.17 |
| **Association Rule Mining** | **52.78** | 25.0 | 19.44 | 2.78 | 36.11 | 16.67 | **44.44** | 2.78 | 33.33 | 11.11 | **52.78** | 2.78 |
| **Time Series Decomposition** | **51.22** | 24.39 | 21.95 | 2.44 | 31.71 | 14.63 | **51.22** | 2.44 | 31.71 | 9.76 | **56.1** | 2.44 |
| **Overall** | **50.29** | 25.43 | 20.0 | 4.29 | 34.43 | 15.86 | **45.57** | 4.14 | 30.29 | 10.71 | **54.71** | 4.29 |

| Task | Coherence | | | | Answers Question Adequately | | | | Plot Conclusion | | | |
|------|------|------|------|------|------|------|------|------|------|------|------|------|
| | *WA* | *WO* | *T* | *N* | *WA* | *WO* | *T* | *N* | *WA* | *WO* | *T* | *N* |
| **Sentiment Analysis** | **51.35** | 24.32 | 21.62 | 2.7 | **40.54** | 27.03 | 29.73 | 2.7 | **37.84** | 24.32 | 35.14 | 2.7 |
| **Basic Data Analysis** | **52.59** | 22.96 | 22.22 | 2.22 | **41.48** | 26.67 | 30.37 | 1.48 | **42.96** | 23.7 | 31.11 | 2.22 |
| **Customer Segmentation** | **52.17** | 26.09 | 19.57 | 2.17 | **41.3** | 28.26 | 28.26 | 2.17 | **39.13** | 23.91 | 34.78 | 2.17 |
| **Association Rule Mining** | **50.0** | 25.0 | 22.22 | 2.78 | **41.67** | 25.0 | 30.56 | 2.78 | **41.67** | 22.22 | 33.33 | 2.78 |
| **Time Series Decomposition** | **51.22** | 24.39 | 21.95 | 2.44 | **41.46** | 24.39 | 31.71 | 2.44 | **41.46** | 21.95 | 34.15 | 2.44 |
| **Overall** | **50.0** | 24.29 | 21.57 | 4.14 | **41.14** | 26.0 | 28.86 | 4.0 | **40.57** | 22.86 | 32.43 | 4.14 |

Table 4: SCORER evaluation comparing the performance of AGENTADA with baseline agents. WA refers to wins by AGENTADA, WO to wins by the baseline agent, T to ties, and N to none. See Tables 21, 22, 23, and 24 for detailed task-level results across all 28 tasks in KAGGLEBENCH.

| Rubric | Rating | w/o skill | Poirot | Pandas | InfiAgent | MetaGPT | GPT-4o |
|--------|--------|------|------|------|------|------|------|
| **Depth of Analysis** | *WA* | 49.12 | 59.73 | 63.88 | 56.74 | 57.91 | 61.77 |
| | *WO* | 28.15 | 19.53 | 12.06 | 22.57 | 21.16 | 16.24 |
| | *T* | 20.78 | 19.48 | 22.87 | 19.46 | 19.66 | 20.70 |
| | *N* | 1.95 | 1.26 | 1.20 | 1.23 | 1.26 | 1.29 |
| **Relevance To Goal** | *WA* | 32.54 | 44.86 | 50.95 | 39.08 | 42.86 | 48.07 |
| | *WO* | 16.31 | 9.52 | 6.82 | 12.89 | 10.52 | 8.44 |
| | *T* | 49.10 | 44.39 | 40.96 | 46.73 | 45.40 | 42.29 |
| | *N* | 2.04 | 1.23 | 1.27 | 1.31 | 1.21 | 1.20 |
| **Persona Consistency** | *WA* | 26.50 | 38.11 | 42.65 | 33.02 | 36.27 | 40.40 |
| | *WO* | 10.05 | 7.41 | 5.08 | 9.70 | 7.58 | 6.42 |
| | *T* | 61.18 | 53.25 | 51.09 | 56.00 | 54.94 | 51.92 |
| | *N* | 2.27 | 1.23 | 1.18 | 1.28 | 1.21 | 1.26 |
| **Coherence** | *WA* | 49.47 | 58.57 | 63.90 | 56.28 | 57.47 | 61.78 |
| | *WO* | 27.18 | 19.49 | 12.00 | 22.30 | 20.86 | 16.71 |
| | *T* | 21.35 | 20.75 | 22.92 | 20.15 | 20.38 | 20.25 |
| | *N* | 1.99 | 1.19 | 1.19 | 1.27 | 1.28 | 1.25 |
| **Plot Conclusion** | *WA* | 40.83 | 51.86 | 56.33 | 49.16 | 50.63 | 53.17 |
| | *WO* | 23.21 | 19.53 | 14.10 | 22.14 | 19.76 | 16.34 |
| | *T* | 34.05 | 27.36 | 28.31 | 27.52 | 28.36 | 29.25 |
| | *N* | 1.92 | 1.25 | 1.26 | 1.18 | 1.26 | 1.24 |

## 6.2 EVALUATION OF AGENTADA'S INSIGHTS VS. OTHER AGENTS

We compared AGENTADA with baseline agents, as shown in Table 4. Across all criteria, AGENTADA consistently outperforms all baselines, confirming the effectiveness of skill-informed analysis. Notably, AGENTADA shows the strongest performance gains over the Pandas agent, with win rates of 63.88% in Depth of Analysis, 63.9% in Coherence, and 56.33% in Plot Conclusion, indicating a clear advantage in generating deeper, clearer, and more structured insights. This gap reflects the limitations of Pandas, which relies on rule-based natural language–to–code translation, compared to AGENTADA's skill-guided code generation. AGENTADA also demonstrates strong performance against powerful agents like GPT-4o and MetaGPT. Though these models are capable of generic reasoning, their lack of analytical skill grounding leads to shallow insights. We also benchmarked AGENTADA against code generation agents like ECA and Evor (Wang et al., 2024; Su et al., 2024b) with details provided in Appendix E and assessed the performance in other benchmarks in Appendix K. Overall, these findings reinforce the value of embedding structured analytical skills into LLM-based data agents.

Table 5: Correctness of insights on KaggleBench QA. Accuracy is exact match with ground truth, while other metrics are rubric scores (1–5) using SCORER.

| Rubric | AgentAda w skill | w/o skill | Poirot | Pandas | InfiAgent | Meta GPT | GPT-4o |
|---|---|---|---|---|---|---|---|
| Accuracy | **90.6** | 82.6 | 81.2 | 80.4 | 84.2 | 85.5 | 78.8 |
| Depth of Analysis | **4.46** | 4.10 | 4.00 | 3.97 | 4.06 | 4.03 | 3.92 |
| Answers Adequately | **4.41** | 4.06 | 3.95 | 3.91 | 4.02 | 3.99 | 3.88 |
| Coherence | **4.48** | 4.09 | 3.97 | 3.94 | 4.03 | 4.00 | 3.90 |
| Relevance to Goal | **4.44** | 4.11 | 3.98 | 3.95 | 4.05 | 4.02 | 3.89 |
| Persona Consistency | **4.42** | 4.07 | 3.96 | 3.93 | 4.04 | 4.01 | 3.87 |

## 6.3 EVALUATING THE FACTUALITY OF THE INSIGHTS

We assess the factual consistency of agent outputs using multiple factuality metrics. Proprietary LLMs such as GPT-5 OpenAI (2025), GPT-4o OpenAI et al. (2024), and LLaMA-3.3-70B Grattafiori et al. (2024) were prompted to rate outputs on a 1–5 scale, where **1** denotes frequent errors and **5** denotes full consistency with the data. In addition, we apply **FactScore** Min et al. (2023), which decomposes outputs into atomic claims and verifies them against ground-truth tables in KaggleBench. In our setup, we follow the original FactScore methodology and use the KaggleBench dataset, consisting of 700 examples across diverse notebooks. For each notebook, we utilize the associated QA list to extract evaluation questions and restrict atomic claims to those grounded in the underlying data tables, excluding suggestions or speculative outputs not present in the source. We compute FactScore with multiple verifiers, including GPT-5, GPT-4o, LLaMA-3.3-70B, and Claude-Opus 4. Table 11 shows that **AgentAda with skills** consistently outperforms its ablation and other baselines, confirming that skill retrieval improves factual grounding.

## 6.4 EVALUATING THE CORRECTNESS OF INSIGHTS

Next, we evaluate the correctness of analysis by AgentADA and compare it with different baselines on KaggleBench. We report accuracy against ground-truth QA and the other rubrics (Depth of Analysis, Answers Adequately, Coherence, Relevance to Goal, Persona Consistency) are rated on a 1–5 scale using SCORER. As shown in Table 5, AgentAda with skill retrieval consistently achieves the highest scores, outperforming both its w/o-skill variant and all other agents. The results show that selecting more appropriate skills leads to stronger analysis and, in turn, more reliable insights.

## 6.5 EVALUATING THE PERFORMANCE OF SKILL MATCHER

To assess the performance of our Hybrid RAG-based skill matcher, we frame it as a ranking task and evaluate how accurately it retrieves relevant skills for each question in KAGGLEBENCH. For each annotated question, the matcher retrieves the top-$k$ skills, which are compared against the ground-truth skills in KAGGLEBENCH. We use Mean Reciprocal Rank (MRR) as our primary metric, measuring the rank position of the first correct skill retrieved. It is defined as $MRR = \frac{1}{N} \sum_{i=1}^{N} \frac{1}{rank_i}$, where $N$ is the total number of queries, and $rank$ represents the rank position of the first correct result for the $i^{th}$ query. We also report Exact Match Accuracy, indicating whether at least one of the retrieved skills matches the ground truth. The matcher achieves high performance, with an **MRR of 0.83 and accuracy of 0.9**, demonstrating its effectiveness in identifying contextually relevant skills.

## 7 CONCLUSION & FUTURE WORKS

We presented AGENTADA, a skill-informed data analytics agent that integrates curated analytical knowledge with LLM capabilities to produce structured, insightful, and goal-aligned analysis. Through extensive evaluation on KAGGLEBENCH, AGENTADA demonstrates significant gains over strong baselines, both in human and LLM-as-a-judge evaluations. Looking ahead, we aim to expand AGENTADA's capabilities beyond structured data analytics, incorporating a more generic skill set for complex tasks and tackling challenges involving unstructured data, multi-table analysis, and large-scale datasets to further enhance its adaptability and real-world applicability.

ETHICS STATEMENT

Our work on AGENTADA does not involve personal or sensitive data, but it raises important considerations. All datasets used are curated from public or sanitized sources, and all human annotators gave informed consent and were fairly compensated. In real-world deployments, however, analytics agents may interact with proprietary or sensitive data, so safeguards such as access control, anonymization, and auditing are essential. Moreover, automatic method selection can inadvertently propagate bias in the data, underscoring the need for fairness audits and transparency in reasoning.

We employed Large Language Models (LLMs) only as writing aids to improve clarity, grammar, and readability of the manuscript. The conceptual framing, methodological design, experimental implementation, analysis, and conclusions remain entirely the work of the authors.

REPRODUCIBILITY STATEMENT

We have taken several steps to ensure reproducibility. The AGENTADA codebase, skill library, and evaluation scripts will be released under a permissive license. KAGGLEBENCH benchmark will also be made publicly available. Implementation details, prompt templates, hyperparameters, and ablation settings are documented in the appendices, and each analytical skill in the library is accompanied by clear input/output documentation and usage examples. Finally, we release baseline agents and evaluation protocols to replicate our experiments and results, making the system transparent and extensible for future research.

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

## A    AGENTADA STATISTICS

Table 6: Summary statistics for KAGGLEBENCH.

| Statistic | Value |
|---|---|
| Total Datasets | 4,304 |
| Average Datasets Per Notebook | 6.15 |
| Total QA Pairs | 6,876 |
| Average Question Token Length | 11.96 |
| Average Answer Token Length | 13.79 |
| Non-null Dataset Descriptions | 526 |
| Average Description Length | 45.56 |
| Notebooks Needing Multiple Files | 187 |

KAGGLEBENCH is a diverse benchmark created based on the notebooks from Kaggle. Table 6 illustrates summary of the statistic in KAGGLEBENCH.

Fig 4 illustrates the domains of the datasets in KAGGLEBENCH. KAGGLEBENCHencompasses 49 distinct domains, with Entertainment and Finance predominating. This predominance reflects the underlying distribution of data analytics datasets on Kaggle. The inclusion of a wide array of domains validates KAGGLEBENCH's utility for diverse data analytics applications.

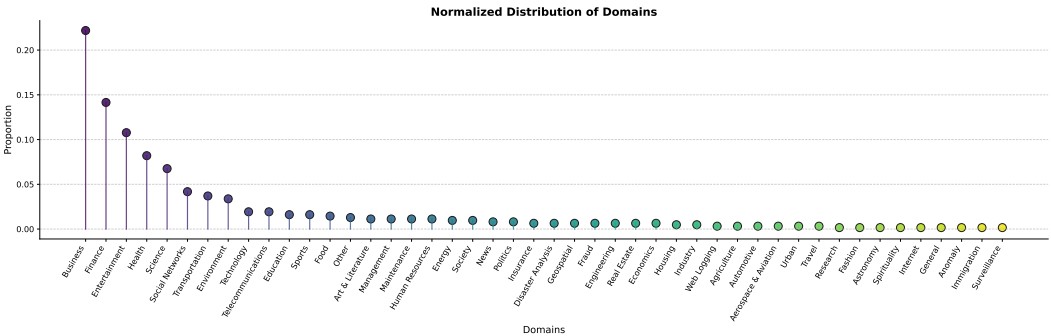

Figure 4: The distribution of domains covered by KAGGLEBENCH

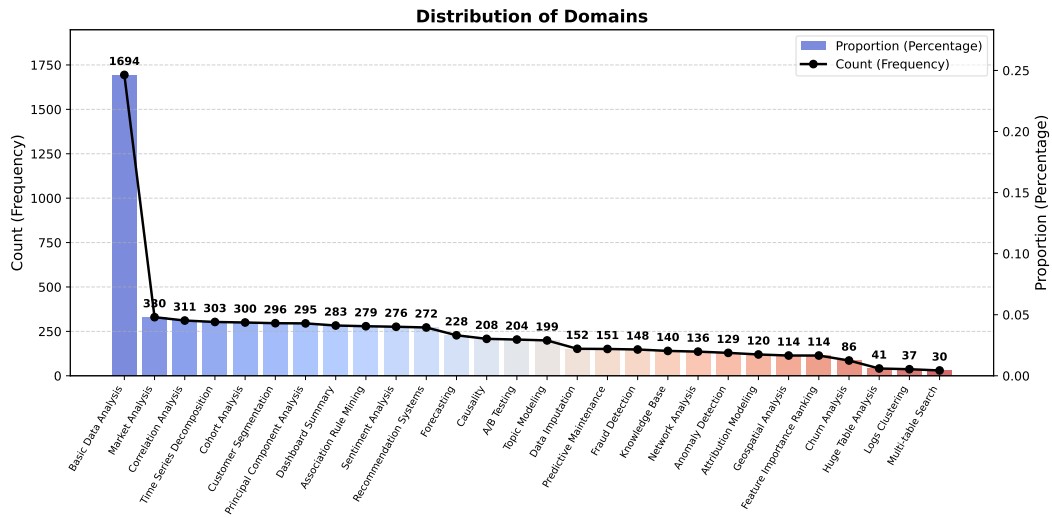

Figure 5: The distribution of tasks covered by KAGGLEBENCH.

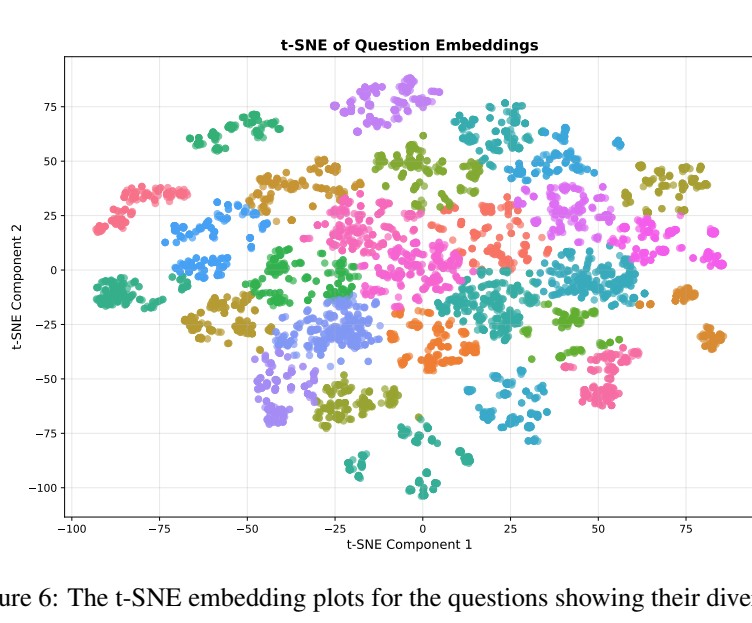

Figure 6: The t-SNE embedding plots for the questions showing their diversity.

In addition to diverse domains, the dataset emphasizes questions that span a variety of tasks. These questions, curated directly from Kaggle notebook cells, cover 28 distinct tasks, as depicted in Fig 5. Notably, the majority focus on Basic Data Analysis, which is expected given its central role in data analytics. Furthermore, we converted the questions into BERT embeddings and applied K-means clustering—with 28 clusters—on the t-SNE projections of these embeddings, as illustrated in Fig 6, to highlight the fact that diversity of questions aligns with the different tasks assigned to them.

## B    SKILL LIBRARY

Table 7 lists the 28 different tasks and the 74 associated skills included in our skill library for AGENTADA, as well as the specific skills required by the tasks in KAGGLEBENCH. This comprehensive set captures the diverse capabilities necessary for effectively solving the wide range of tasks represented in the dataset.

## C    HUMAN EVALUATION PLATFORM

Human evaluation was conducted using Gradio app, an interactive tool that simplifies the evaluation process with its intuitive interface while enabling real-time feedback and iterative improvements for a comprehensive, user-centered assessment of our model's performance. Following are the 6 steps outlining the procedure of human evaluation (as illustrated in Figure 7):

1. Choose 'User designation' from the drop-down list.

2. 'Dataset ID' is a slider which shows the dataset index that is being evaluating currently. 'Dataset Information' gives detailed description about the dataset. This is very useful for evaluators if they loose connection in between or would want to get back after taking a break.

3. 'Question Index' shows the index of the question which is being evaluated. Each 'Dataset ID' has 3 questions with a unique index for each. Similar to (2),this slider is quite resourceful for evaluators if they loose connection in between or would want to get back after taking a break.

4. The 2 models are represented as 'A' and 'B'. One of them uses the skill and the other doesn't use(this is randomly chosen each time to keep it unbiased). Each of these model shows the plot and answer corresponding to the question.

5. The goal defines the primary objective—what the project aims to achieve using the dataset. This could involve uncovering patterns, solving a specific problem, making predictions, or

Table 7: Tasks and corresponding skills available for AGENTADA and KAGGLEBENCH.

| TNo | Task | Skills |
|-----|------|--------|
| 1 | Sentiment Analysis | BERT, LSTM, Naive Bayes |
| 2 | A/B Testing | Student's T-Test, Multi-Armed Bandit |
| 3 | Forecasting | ARIMA, Prophet, LSTM |
| 4 | Fraud Detection | Random Forest, Isolation Forest, Neural Networks |
| 5 | Recommendation Systems | Collaborative Filtering, Matrix Factorization, Deep Neural Networks |
| 6 | Churn Analysis | Gradient Boosting Machines, Random Forest |
| 7 | Customer Segmentation | K-means Clustering, RFM Analysis, Hierarchical Clustering |
| 8 | Network Analysis | PageRank, Louvain Method, Betweenness Centrality |
| 9 | Association Rule Mining | Apriori Algorithm, FP-Growth, ECLAT |
| 10 | Dashboard Summary | KPI Analysis, Interactive Visualization, Statistical Aggregation |
| 11 | Predictive Maintenance | LSTM, Random Forest, Gradient Boosting Machines |
| 12 | Cohort Analysis | Retention Analysis, Sequential Pattern Mining |
| 13 | Attribution Modeling | Markov Chains, Shapley Value Attribution, Multi-Touch Attribution |
| 14 | Anomaly Detection | Isolation Forest, Local Outlier Factor, One-Class SVM |
| 15 | Feature Importance Ranking | Random Forest Importance, SHAP Values, LASSO Regularization |
| 16 | Geospatial Analysis | Kernel Density Estimation, Spatial Autocorrelation, DBSCAN for Spatial Clustering |
| 17 | Causality | Structural Equation Modeling, Granger Causality, Propensity Score Matching |
| 18 | Logs Clustering | DBSCAN, LogCluster, Word2Vec with K-means |
| 19 | Time Series Decomposition | Seasonal-Trend Decomposition, Wavelet Decomposition |
| 20 | Principal Component Analysis | SVD, Eigenvalue Decomposition, Kernel PCA |
| 21 | Correlation Analysis | Pearson Correlation, Spearman Correlation, Kendall's Tau |
| 22 | Knowledge Base | BERT, Latent Semantic Analysis, PageRank |
| 23 | Multi-table Search | B+ Tree Indexing, Hash Join Algorithms, Bitmap Indexing |
| 24 | Huge Table Analysis | MapReduce, Columnar Storage Processing, Approximate Query Processing |
| 25 | Topic Modeling | Latent Dirichlet Allocation, Non-negative Matrix Factorization, Hierarchical Dirichlet Process |
| 26 | Market Analysis | Time Series Analysis, Market Basket Analysis, K-Means Segmentation |
| 27 | Data Imputation | MICE, KNN Imputation, Random Forest Imputation |
| 28 | Basic Data Analysis | Basic Data Analysis |

informing strategic decisions. On the other hand, the persona represents a realistic profile of the intended user or stakeholder who will interact with the data or benefit from the insights. It includes their background, expertise, objectives, and challenges. Together, the goal and persona ensure that the analysis remains focused, relevant, and tailored to deliver meaningful value to the right audience.

6. A total of 6 Rubrics have been used for this evaluation study. They are as follows:

   a **Depth of Analysis**: Evaluates whether the response goes beyond surface-level descriptions to uncover non-trivial patterns, relationships, or trends in the data. A high score indicates multi-step reasoning and interpretation that adds real analytical value.

   b **Relevance to Goal**: Assesses how well the response remains aligned with the stated analytical objective or research question. Strong responses avoid unnecessary detours and keep the analysis tightly focused on achieving the intended goal.

   c **Persona Consistency**: Measures whether the response is consistent with the intended persona's expertise, perspective, and communication style. For instance, a "business analyst" persona should emphasize actionable insights, while a "data scientist" persona should highlight methodological rigor.

d **Coherence**: Examines the clarity, logical structure, and internal consistency of the response. Higher scores indicate that arguments, evidence, and conclusions are presented in a well-organized and easy-to-follow manner.

e **Answering Question Adequately**: Determines whether the response fully and correctly addresses the question posed. Partial answers, misinterpretations, or omissions lower the score, while comprehensive and precise responses score higher.

f **Plot Conclusion Quality**: Evaluates the correctness and clarity of conclusions drawn from plots or visualizations. Strong responses explicitly connect the visual evidence to the broader analysis, offering a clear and accurate takeaway rather than vague or generic statements.

A "comment box" has been provided which can be used to give an explanation/reason for the choice of answer.

7. At the end, after making choices and providing comments; Click on 'Submit rubrics' to save the evaluation responses in JSON file (Figure 8)! 'Previous' goes to the previous question and 'Next' takes you to the next question. Clicking on 'Submit rubrics' is necessary so as to save the evaluation.

## D  HUMAN EVALUATION STATISTICS AND DETAILS

We recruited 30 participants through a Google Form, which included task instructions and an estimated completion time based on our pilot study (1.5–2.5 minutes). Among the participants, 21 were male and 9 were female. As detailed in Appendix C, we also recorded each participant's professional designation. Figure 9 shows the distribution of evaluator expertise.

| Depth of Analysis | Relevance to Goal | Persona Consistency | Coherence | Answers Question Adequately | Plot Conclusion |
|---|---|---|---|---|---|
| 0.8842 | 0.8297 | 0.8431 | 0.7658 | 0.8274 | 0.8765 |

Table 8: Fleiss' Kappa scores for inter-annotator agreement across evaluation rubrics.

## E  ABLATION: COMPARISON WITH OTHER CODE GENERATORS

We next compare AgentAda with two representative code generation baselines ECA: (Wang et al., 2024) and Data Interpreter (Hong et al., 2024). Both approaches rely on invoking existing tools or examples but lack the structured, skill-based generation that distinguishes AgentAda.

**ECA:** The prompt used for ECA is given in Prompt 1. ECA generates code by directly calling statistical or machine learning tools (e.g., ARIMA, Granger causality tests) without constructing full pipelines. As a result, it frequently misses prerequisite steps such as preprocessing, diagnostics, or lag order selection, leading to incomplete or misleading analyses. For example, in the Granger causality task in the Table 9, ECA reduces the problem to testing a variable against its own lag, producing an ninformative setup. This reflects a fundamental limitation, ECA is a tool invoker rather than an analysis engine.

**Data Interpreter (MetaGPT)**. Data Interpreter generates code using few-shot prompting, retrieving semantically similar examples and directly stitching together function calls. While this can yield runnable code, it often fails to produce task-specific pipelines aligned with the analytical goal, since it does not incorporate user intent or persona. By contrast, AgentAda's access to skill library act as structured blueprints that allow AgentAda to generate code from scratch, personalized to the goal and persona.

**AgentAda**. The key distinction is that AgentAda is not a code generator but a full analysis engine. By grounding generation in skills, it produces pipelines entirely from scratch using skills from the skill library that encapsulate step-by-step workflow. Crucially, this is not borrowed or stitched-together code, but code that is personalized to the specific task which ensures necessary preprocessing, diagnostics, modeling, and evaluation steps are correctly integrated.

We also conducted a quantitative comparison of AgentAda, ECA, and EvoR on all QA pairs of KaggleBench. Accuracy was measured directly against the benchmark QA answers, while rubric-based

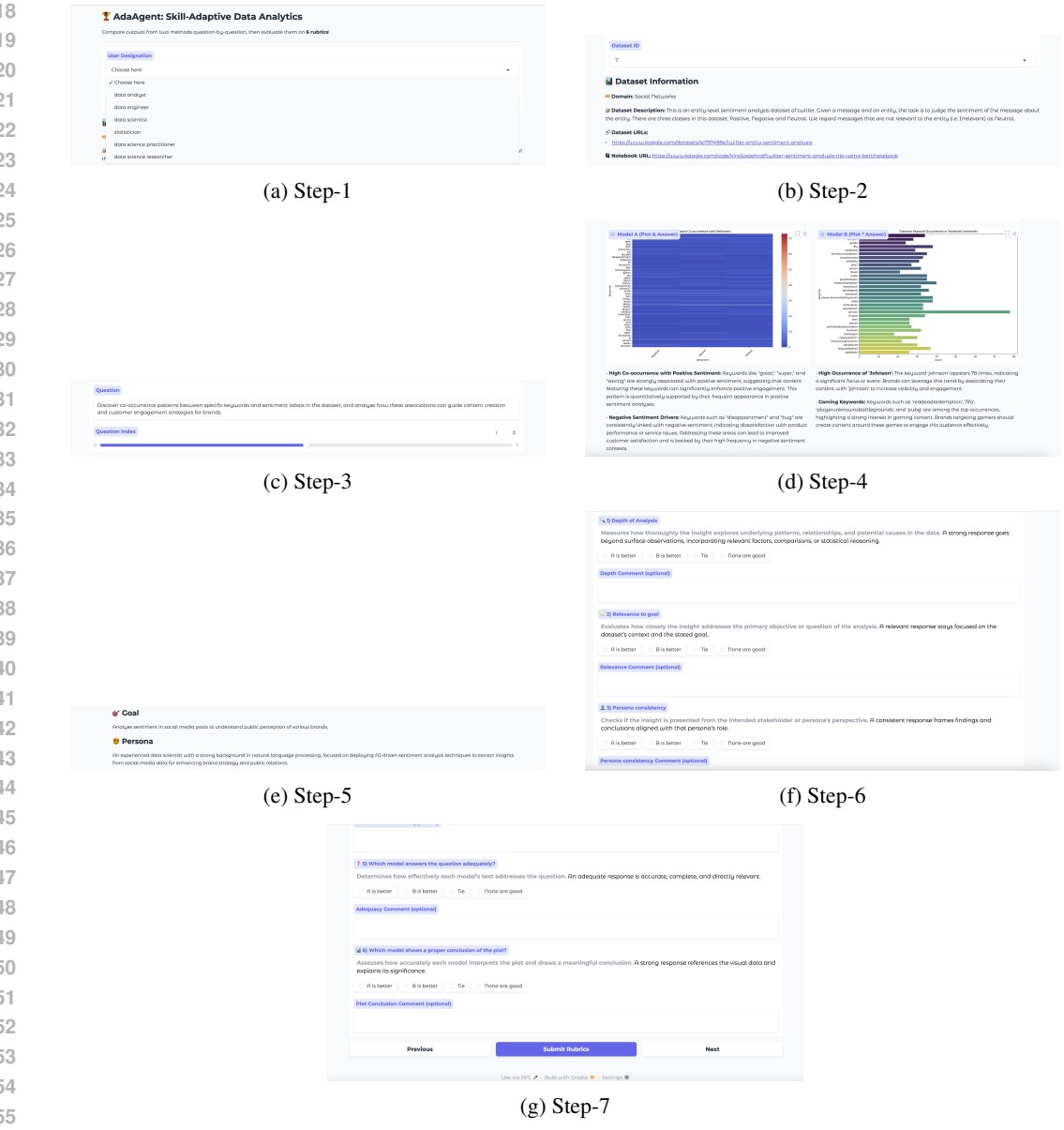

(a) Step-1

(b) Step-2

(c) Step-3

(d) Step-4

(e) Step-5

(f) Step-6

(g) Step-7

Figure 7: Human Evaluation Platform Step-by-Step Workflow.

dimensions were judged on a 1–5 scale using GPT-4o tuned with SCORER. The rubrics evaluate depth of analysis, adequacy of answers, coherence, relevance to goal, and persona consistency.

As shown in Table 10, AgentAda achieves the highest accuracy (90.6%) and outperforms both ECA and EvoR (Su et al., 2024b) across all rubric dimensions. These results are consistent with the above qualitative case study on Granger causality where AgentAda generated a complete and methodologically sound pipeline. Together, the quantitative and qualitative evidence highlight AgentAda's advantage as a full analysis engine that generates task-specific, coherent, and goal-aligned analyses, unlike baselines that rely on tool invocation or code stitching.

## F    EVALUATING THE FACTUALITY OF THE INSIGHTS

We further assess the factual consistency of agent outputs using multiple factuality metrics. For large proprietary models such as GPT-5 OpenAI (2025), GPT-4o OpenAI et al. (2024) LLaMA-3.3-70B

```
Archive > {} 203_1_20250322_233954.json > ...
  1  {
  2      "dataset_id": 203,
  3      "question_idx": 1,
  4      "timestamp": "20250322_233954",
  5      "designation": "statistician",
  6      "user_id": "f23c7b25-5f8c-4acb-917c-a40adfe68c86",
  7      "rubrics": {
  8          "depth_of_analysis": {
  9              "selection": "A is better",
 10              "comment": ""
 11          },
 12          "relevance_to_goal": {
 13              "selection": "Tie",
 14              "comment": ""
 15          },
 16          "persona_consistency": {
 17              "selection": "Tie",
 18              "comment": ""
 19          },
 20          "coherence": {
 21              "selection": "B is better",
 22              "comment": ""
 23          },
 24          "answers_question_adequately": {
 25              "selection": "Tie",
 26              "comment": ""
 27          },
 28          "plot_conclusion": {
 29              "selection": "A is better",
 30              "comment": ""
 31          }
 32      },
 33      "model_a": {
 34          "exp_group": "SuperBatch3/insights_W_batch_1",
 35          "hash": "W_1",
 36          "skill": "RFMAnalysis",
 37          "output": "* **High Churn Rates in Short Tenure Cohorts:** Cohorts with tenures of 0-1, 0-2, and 0-3 exhibit churn rates reaching 1.0, indicating a critical need for early e
 38      },
 39      "model_b": {
 40          "exp_group": "SuperBatch3/insights_Wo_batch_1",
 41          "hash": "Wo_1",
 42          "skill": "RFMAnalysis",
 43          "output": "* **High Churn Rate in Early Tenure with Few Products:** The cohort '0-3' exhibits a maximum churn rate of 1.0, indicating that customers with very short tenure a
 44      }
 45  }
```

Figure 8: Human evaluation result file (in JSON).

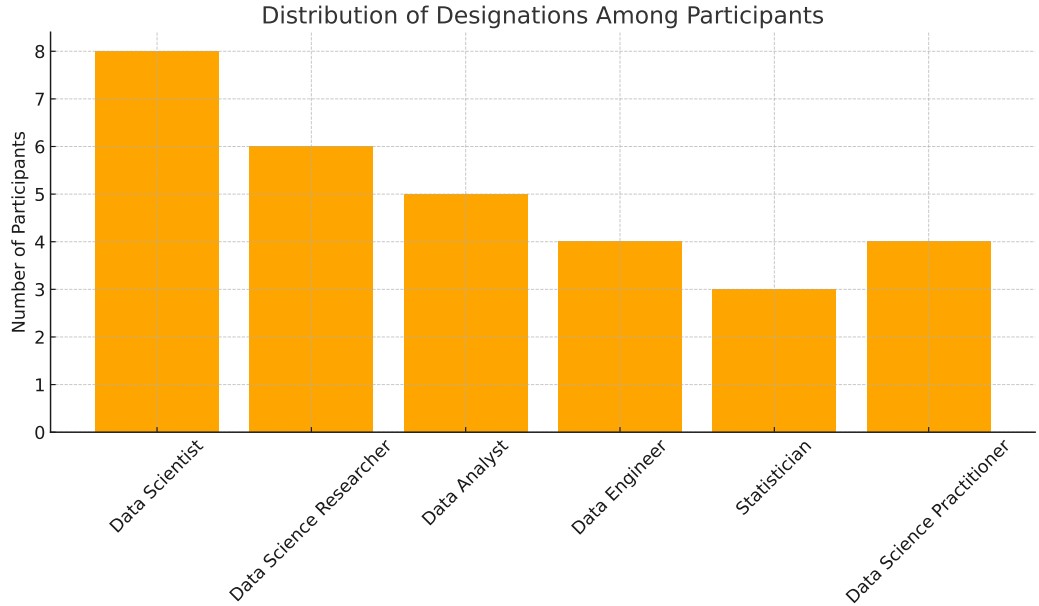

Figure 9: Distribution of expertise of the human evaluators.

Grattafiori et al. (2024), we directly query the models to provide a factuality rating on a 1–5 scale, where **1** indicates that the output contains many factual errors and is mostly ungrounded, while **5** indicates that the output is fully consistent with the provided data and free of factual mistakes. In addition, we employ **FactScore** Min et al. (2023), which evaluates factuality by decomposing model outputs into atomic claims and verifying them against ground-truth sources. In our setup, we follow the original FactScore methodology and use the KaggleBench dataset, consisting of 700 examples across diverse notebooks. For each notebook, we utilize the associated QA list to extract evaluation questions and restrict atomic claims to those grounded in the underlying data tables,

---

**🗄 ECA**

You have access to the following tools: [1] adfuller: Performs the Augmented Dickey-Fuller test for stationarity of a time series. Arguments: x (array-like), maxlag (int, optional), regression (str), autolag (str), store (bool), regresults (bool) Returns test statistic, p-value, used lag, number of observations, critical values, and optional info. Signature: adfuller(x: ArrayLike, maxlag: int = None, regression: str = 'c', autolag: str = 'AIC', store: bool = False, regresults: bool = False) -> Tuple

[2] kpss: Performs the Kwiatkowski-Phillips-Schmidt-Shin test for stationarity. Arguments: x (array-like), regression (str), nlags (str or int), store (bool) Returns test statistic, p-value, lags used, and critical values. Signature: kpss(x: ArrayLike, regression: str = 'c', nlags: Union[str, int] = 'auto', store: bool = False) -> Tuple

[3] acf: Computes the autocorrelation function of a time series. Arguments: x (array-like), nlags (int), alpha (float), fft (bool), missing (str) Returns autocorrelations and optionally confidence intervals. Signature: acf(x: ArrayLike, nlags: int = 40, alpha: float = None, fft: bool = True, missing: str = 'none') -> Union[np.ndarray, Tuple[np.ndarray, np.ndarray]]

[4] pacf: Computes the partial autocorrelation function of a time series. Arguments: x (array-like), nlags (int), method (str), alpha (float) Returns partial autocorrelations and optionally confidence intervals. Signature: pacf(x: ArrayLike, nlags: int = 40, method: str = 'ywunbiased', alpha: float = None) -> Union[np.ndarray, Tuple[np.ndarray, np.ndarray]]

[5] ccf: Computes the cross-correlation function between two time series. Arguments: x (array-like), y (array-like) Returns cross-correlations. Signature: ccf(x: ArrayLike, y: ArrayLike) -> np.ndarray

[6] grangercausalitytests: Performs Granger causality tests for all lags up to a specified maxlag. Arguments: x (array-like), maxlag (int), addconst (bool), verbose (bool) Returns test results dictionary for each lag. Signature: grangercausalitytests(x: ArrayLike, maxlag: int, addconst: bool = True, verbose: bool = True) -> Dict

[7] arma_order_select_ic: Selects optimal AR and MA order using AIC/BIC. Arguments: y (array-like), max_ar (int), max_ma (int), ic (str), trend (str) Returns dictionary with selected order and full results. Signature: arma_order_select_ic(y: ArrayLike, max_ar: int, max_ma: int, ic: str = 'aic', trend: str = 'c') -> Dict

[8] lagmat: Creates a 2D array of lagged versions of a time series. Arguments: x (array-like), maxlag (int), trim (str), original (str) Returns lagged matrix. Signature: lagmat(x: ArrayLike, maxlag: int, trim: str = 'forward', original: str = 'ex') -> np.ndarray

[9] acovf: Computes autocovariance function of a time series. Arguments: x (array-like), unbiased (bool), demean (bool), nlag (int), fft (bool), missing (str) Returns autocovariances. Signature: acovf(x: ArrayLike, unbiased: bool = False, demean: bool = True, nlag: int = None, fft: bool = False, missing: str = 'none') -> np.ndarray

Now, let's get started! Instruction: You are given a dataset with the following columns: Month: A numeric representation of time Passengers: Number of airline passengers in that month Answer this question about the data: Does an increase in passenger volume during one month lead to higher passenger counts in subsequent months due to seasonal travel momentum or compounding economic effects? You can optionally express your thoughts using natural language before your action. For example, 'Thought: I want to use tool_name to do something. Action: <your action to call tool_name> End Action'. Note that your output should always contain either 'Action:' or 'Answer:', but not both. When you are done, output the result using 'Answer: your answer'

Prompt 1: Prompt given to GPT-4o using the **ECA** framework.

excluding suggestions or speculative outputs not present in the source. We compute FactScore with multiple verifiers, including GPT-5, GPT-4o, LLaMA-3.3-70B, and Claude-Opus 4.

Table 11 reports the results. We observe that **AgentAda with skills** consistently outperforms its ablation and other baselines across all factuality metrics, confirming that skills contribute to generating more accurate, data-grounded insights.

# G   ABLATION: INFLUENCE OF GOAL AND PERSONA

AGENTADA insights can be tailored to the user's goal and persona to produce specific types of analysis. However, as shown in Table 4, other agents like the Pandas agent—which does not receive goal or persona inputs—still perform well in some cases. This suggests that such information might be inferred from the structure of the data itself. As a result, we removed the "goal" and "persona" inputs from our pipeline to examine their impact. We evaluated both versions of the model on the same 100 datasets from KAGGLEBENCH used in the human evaluation, using the insight-wise SCORER metric for comparison (the question generated in the pipelines are different so we need to do insight-wise comparison).

As shown in Table 12, both goal and persona influence the quality of the final insights across all evaluation rubrics. However, most of the impact comes from the goal, followed by the persona, while differences in the other rubrics are relatively minor. This may be because goal and persona help align the model's chain of thought during analysis, leading to improved results. Between the two, the goal has a stronger effect on the goal relevance rubric than the persona does on persona consistency. This could be because personas are typically more generic and less tied to the specific type of analysis or skills required. Additionally, while the persona is only used during question generation, the goal is used in both question and insight generation, making it more influential on the final outputs.

# H    ABLATION: ALIGNMENT OF SCORER WITH HUMAN FEEDBACK

In this section we evaluate how well different evaluators align with human judgment when comparing two variants of our system: AgentAda with skills versus AgentAda without skills. Each evaluator must decide which variant produces better insights on KaggleBench datasets, or mark the outcome as a tie or none. We consider four evaluators: SCORER, Human Evaluation, G-Eval, and LLaMA-Eval. Note that, the data used to train SCORER is disjoint from the validation set used for this experiment, ensuring no leakage or bias. As shown in Table 13, SCORER aligns much more closely with human evaluation than G-Eval and LLaMA-Eval. This is expected, as SCORER is explicitly optimized for human alignment.

# I    PROMPTS

## I.1    KAGGLEBENCH PROMPTS

Prompt 2 and Prompt 3 illustrates the prompts used for generating the question answer pairs and goal & persona for KAGGLEBENCH respectively.

---

**⛁ KAGGLEBENCH QA Pair Generation**

You have the following dataset:
{dataset_summary}

The following are the notebook cells provided to give context and examples of possible data analytics tasks:
{cells}

Relevant data analytics tasks include:
- {task_1}
- {task_2}
- ...
- {task_n}

{skills_section}

- {skill_1}
- {skill_2}
- ...
- {skill_n}

Instructions 1. Generate a list of **questions and answers** related to **data analytics tasks** that can be performed on the dataset.
2. Each question should:
- Focus on analyzing or gaining insights from the dataset itself (not the notebook).
- Be framed from the perspective of someone analyzing the dataset directly.
- Include the specific data analytics task and skill required to answer it.
3. Use the notebook cells as inspiration for possible types of analytics, but do not ask questions directly about the notebook's implementation.
4. For each generated question and answer, include:
- The cell numbers that informed the question (if any).
- The data analytics task and skill required.
5. Your answers to the question should only come from the cells (usually the output cells or the markdown cells). Your answer should not be out of the given cell context.
6. First choose the task, and then choose the skill needed to answer the question based on the list of skills for that specific task.

Expected Output [IMPORTANT]
1. The question should be about data. Meaning, if a person sees the data, what analytical question might they ask. The cells given from the notebook are only giving ideas about the type of analytics that can be done.
2. The answer should be derived from the cells (usually outputs). No analysis should be done outside the given cells. The cells are the only source of information for questions and answers.
3. Include different question types, from basic data analysis questions for understanding the data to detailed questions like asking about the number of clusters in the data, which comes from doing a clustering (this has been done in the notebook).
4. The task and skill should be selected from the list of tasks and skills provided.

---

Prompt 2: Prompt to GPT-4o to generate QA pairs from each notebook. The answers were validated with RAG-Token Model as describe in Section 3.

> **⛁ KAGGLEBENCH Goal and Persona extraction in KAGGLEBENCH**
>
> You are an expert data analyst who has just finished working with a dataset and the associated notebook content. I will provide you with: 1. A dataset summary: This is a textual description of the dataset, including its columns, values, features, and overall purpose. 2. Questions: A list of dataset-specific questions reflecting insights derived from the dataset or the analysis described in the notebook. Your task is to analyze these inputs and generate a JSON response containing: - Goal: The primary goal of the analysis based on the dataset summary and notebook content. Describe what the analysis aims to achieve, without the models and analyses used as that should be what the analytics agent should figure out. - Persona: A detailed description of the person conducting the analysis. Include information such as their profession, expertise level, goals, and interests.
>
> For example: 'A marketing analyst with 5 years of experience in e-commerce, focused on understanding customer behavior and optimizing marketing strategies for revenue growth.'
>
> Instructions: - The goal should be a one line and short description of what is the purpose of the analysis. - The goal should be short and be "what" is the goal instead of "how" it is done.
>
> - Goal is the the goal that a data analyst would have without telling him/her which methods to use.
>
> - The persona should be detailed and should be a persona of a data analyst who is analyzing the data.
>
> Ensure your response is concise, well-structured, and grounded in the provided inputs. Generate the output as a valid JSON object. You should provide only a JSON file as the output. No additional information is needed.

Prompt 3: Prompt to GPT-4o to extract goal and persona from each notebook.

## I.2 DUAL STAGE ADVANCED QUESTION GENERATION PROMPTS

To generate dataset-specific questions, we initially prompted GPT-4o-mini (OpenAI et al., 2024) to produce five basic questions that aid data analysts in understanding a dataset. The prompt (see Prompt 4) accepts input parameters such as the dataset's analysis goal, the analyst's persona, the names and data types of the dataframe columns, and the dataframe head. An output template is also provided to ensure consistent formatting. The primary objective of this prompt is to generate five questions that offer fundamental insights into the dataset.

Subsequently, these basic questions—along with the original input information—are fed into a specialized advanced question generation prompt (see Prompt 5). This prompt, also leveraging GPT-4o-mini (OpenAI et al., 2024), is designed to generate skill-oriented questions. We supply an output format template that organizes the output into distinct task and question components for consistency. The main focus of this advanced prompt is to produce questions that require the advanced analytical skills defined in our skill library, thereby uncovering deeper insights into the dataset and yielding more actionable results.

We also explored a single-prompt approach for the question generation pipeline. The prompt (see Prompt 6) accepts the same inputs as our advanced question generation prompt, with the exception of the basic generated questions. However, this approach yielded questions that were either overly similar or did not align with the advanced analytical requirements we aimed to address.

Fig 10 and 11 show examples of the basic and advanced question generated by the dual stage pipeline. While Fig 12 show an example for the questions generated by the single stage pipeline. It is evident from the questions that the advanced questions generated by our dual stage pipeline are more complete and cover a diverse range of skills that could help in uncovering patterns in the dataframe that the single stage pipeline would not. Hence, necessitating the need for our dual stage pipeline.

## I.3 CATEGORY PREDICTION PROMPTS

To guide GPT-4o in predicting high-level insight themes for a given dataset, we design a structured prompt that provides the model with (I) the dataset description, (II) the overall analysis goal, and (III) the list of generated analytical questions. The goal of the prompt is to predict exactly three distinct, meaningful categories that are broad enough to group multiple related insights but specific enough to remain actionable and aligned with the context of the analysis. The prompt (see Prompt 7) emphasizes:

---

**Example of Basic Questions Generated by Dual Stage Pipeline:**

1. What is the correlation between tenure and customer churn, and how does it vary across different customer demographics such as gender and SeniorCitizen status?

2. How do different InternetService types (DSL, Fiber optic, No) impact the likelihood of customer churn, and what additional services (like OnlineSecurity or TechSupport) are most associated with retention?

3. What role do payment methods play in customer churn rates, and are there specific payment methods that correlate with higher retention?

4. How do MonthlyCharges and TotalCharges relate to customer churn, and are there specific thresholds that indicate a higher risk of churn?

5. What patterns can be identified in the combination of services used (e.g., PhoneService, MultipleLines, StreamingTV) that correlate with higher customer satisfaction and lower churn rates?

---

Figure 10: Example of Basic Questions Generated by Dual Stage Pipeline for dataset id 201.

---

**Example of Advanced Questions Generated by Dual Stage Pipeline:**

1. **Churn Analysis**: Develop a predictive model to identify high-risk customer segments based on their service usage patterns and demographic information, and suggest targeted retention strategies that align with the goal of reducing churn in the telecommunications sector.

2. **Cohort Analysis**: Analyze customer behavior over time by grouping customers based on their tenure and service usage, and identify trends that correlate with churn rates, providing insights for tailored retention initiatives that resonate with the persona's expertise.

3. **Association Rule Mining**: Explore the relationships between different service combinations (e.g., InternetService, OnlineSecurity, TechSupport) and churn rates to uncover patterns that can inform service bundling strategies aimed at enhancing customer loyalty.

4. **A/B Testing**: Design an experiment to test the effectiveness of different customer engagement strategies (e.g., personalized offers vs. standard promotions) on reducing churn, and analyze the results to determine which approach yields better retention outcomes.

5. **Network Analysis**: Investigate the interactions between customer service usage and churn by mapping out the relationships between different services and customer demographics, identifying key nodes that could be targeted for retention efforts to improve overall customer satisfaction.

---

Figure 11: Example of Advanced Questions Generated by Dual Stage Pipeline for dataset id 201.

- Avoiding generic or overly broad categories.

- Ensuring non-overlapping, interpretable groupings.

- Aligning categories with the dataset and goal.

Figure 13 an example of the predicted categories for the advanced questions.

### I.4 SKILL MATCHER PROMPTS

To identify the most relevant skills for a given question, we prompt GPT-4o (OpenAI et al., 2024) with the question and a list of all available skills in the library. The prompt asks the model to rank the top three skills based on their usefulness in answering the question. We also provide a structured output template to ensure consistency in formatting. Refer to Prompt 8 for more details.

> **Example of Questions Generated by Single Stage Pipeline:**
>
> 1. **Churn Analysis**: Utilize logistic regression to identify the key factors influencing customer churn, and quantify the impact of each factor on the likelihood of churn, providing actionable insights for retention strategies tailored to the telecommunications sector.
>
> 2. **Customer Segmentation**: Implement k-means clustering to segment customers based on their service usage patterns and demographic characteristics, and analyze how these segments correlate with churn rates to develop targeted retention campaigns.
>
> 3. **Cohort Analysis**: Conduct a cohort analysis to track the retention rates of customers who signed up under different contract types (e.g., month-to-month vs. one year) over time, and assess how these patterns inform strategies for improving customer loyalty.
>
> 4. **Predictive Maintenance**: Develop a predictive model using decision trees to forecast potential churn based on customer behavior and service usage metrics, and evaluate the model's effectiveness in identifying at-risk customers for proactive retention efforts.
>
> 5. **Feature Importance Ranking**: Apply random forest feature importance analysis to rank the variables that most significantly contribute to customer churn, and discuss how these insights can guide the development of personalized customer engagement strategies to enhance satisfaction and reduce churn.

Figure 12: Example of Questions Generated by Single Stage Pipeline for dataset id 201.

---

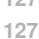 **Basic Data Analytics Questions**

You are an AI assistant specializing in data analysis.
I have a dataset with the following details:

Columns: {columns}
Data Types: {data_types}
Sample Data: {sample_data}
Goal: {goal}
Persona: {persona}

Based on this information, generate five insightful questions that a data analyst in this persona would ask or seek to answer when exploring the dataset.

The questions should be relevant to the dataset's structure and align with the stated goal of the analysis.

Make sure that all the questions are returned as a list named generated_questions The generation format should be:

generated_questions = [question_1, question_2, ..., question_5]

---

Prompt 4: Prompt for Basic Data Analytics Question Generation.

## I.5 CODE GENERATION PROMPTS

To generate the required plot for answering a question, we prompt GPT-4o (OpenAI et al., 2024) using the question and a summary of the selected skill. The prompt (see Prompt 9) is responsible for generating both the code and key statistics about the dataset. It emphasizes structured code generation, producing code that encompasses data preparation, skill application, visualization, computation of key statistics, and adherence to best coding practices.

To ensure that the generated code utilizes the required skill, we pass the code along with our skill list to GPT-4o for verification. This check is performed using Prompt 10.

---

**📚 Advanced Data Analytics Question**

You are an AI assistant specializing in data analysis. I have a dataset with the following details:
Columns: {columns}
Data Types: {data_types}
Sample Data: {sample_data}
Goal: {goal}
Persona: {persona}
Additionally, I have already generated these "basic questions" that a data analyst might ask when exploring this dataset: {generated_basic_questions}
Now, using the provided dataset information, these basic questions, and the goal and persona as guiding principles, "generate {num_questions} additional advanced and diverse questions that require specialized analytical techniques" to answer.
Requirements for the "Advanced Questions":
**Goal Alignment**: Each question must directly contribute to achieving the stated goal of the analysis.
**Persona Relevance**: The complexity and focus of the questions should match the persona's expertise and domain.
**Higher Complexity**: Questions should require deeper analytical skills, making them significantly more advanced than the basic ones.
**Skill-Based**: Each question should necessitate the use of exactly one skill from the following skill list: {skill_list}
-**Implicit Skill Usage**: The skill name must not be directly mentioned in the question.
-**Diverse Techniques**: Ensure a variety of skills are used across the five questions, avoiding redundancy.
Before finalizing a question, **internally reason** if GPT-4o can answer this question using basic reasoning or common-sense knowledge?
- If **yes**, reject the question and generate a more advanced one.
- If **no**, proceed.
Format each question on a new line, and pair it with its corresponding task name, like this:
1. [Task Name] - Question
2. [Task Name] - Question
...
Starting from 1 and ending at {num_questions}...
For example:
1. [**Forecasting**] - Using time series decomposition, predict the seasonal trends in customer engagement over the next 12 months, specifically focusing on how these trends align with the goal of increasing user retention for the persona of a subscription-based business.
2. [**Anomaly Detection**] - Identify unusual patterns in user behavior that may indicate fraudulent activity, and propose methods to mitigate these risks, ensuring the solutions align with the goal of reducing fraud for the persona of a financial services provider.
3. [**Customer Segmentation**] - Apply clustering algorithms to segment customers based on purchasing behavior and sentiment analysis, and recommend targeted marketing strategies for each segment, ensuring the recommendations align with the goal of increasing sales for the persona of an e-commerce platform.
4. [**Causality**] - Investigate the causal relationship between marketing spend and customer conversion rates, controlling for external factors such as seasonality and economic conditions, and provide insights that align with the goal of optimizing marketing ROI for the persona of a digital marketing agency.
5. [**Feature Importance Ranking**] - Rank the most influential features in predicting customer churn using SHAP values, and explain how these features impact retention strategies, ensuring the analysis aligns with the goal of reducing churn for the persona of a telecom company.

Prompt 5: Advanced Question Generation Prompt.

---

**Example Insight Categories:**

1. **Customer Segmentation and Risk Profiling**
   This category will encompass insights related to identifying high-risk customer segments based on service usage patterns and demographic information. It will focus on understanding which customer groups are most likely to churn and why, allowing for targeted retention strategies.

2. **Service Usage Patterns and Churn Correlation**
   This category will capture insights derived from analyzing the relationships between different service combinations and churn rates. It will highlight patterns and trends in service usage that correlate with customer churn, informing strategies for service bundling and customer engagement.

3. **Retention Strategy Effectiveness**
   This category will include insights from experiments and analyses designed to test and evaluate the effectiveness of various customer engagement strategies. It will focus on determining which approaches, such as personalized offers or standard promotions, are most successful in reducing churn and improving customer retention.

Figure 13: Example predicted insight categories for sentiment analysis dataset.

> **📚 Single Stage Question Generation**
>
> Given a dataset with the following characteristics:
> Columns: {columns}
> Data Types: {data_types}
> Sample Data: {sample_data}
> Additionally, consider the following project goal and persona:
> - Goal: {goal}
> - Persona: {persona}
> Generate {num_questions} specific, advanced, and diverse quantitative data analytics questions that could be answered using this dataset. Ensure that the questions:
> 1. **Pertain to the Goal and Persona**: Each question must directly relate to the provided goal and persona. Avoid generating questions that deviate from the context of the goal or persona.
> 2. **Are Diverse and Varied**: The questions should cover a wide range of aspects of the dataset, including but not limited to trends, relationships, anomalies, and actionable insights. Ensure no single area is overrepresented.
> 3. **Are Advanced**: The questions should require deeper analytical thinking, such as multivariate analysis, predictive modeling, or advanced statistical techniques. Avoid basic or superficial questions.
> Each question should be paired with the relevant task name from the following list: {skill_list}
> Format each question on a new line, and pair it with its corresponding task name, like this:
> 1. [Task Name] - Question
> 2. [Task Name] - Question
> ...
> Starting from 1 and ending at {num_questions}...
> For example:
> 1. [Forecasting] - Using time series decomposition, predict the seasonal trends in customer engagement over the next 12 months, specifically focusing on how these trends align with the goal of increasing user retention for the persona of a subscription-based business.
> 2. [Anomaly Detection] - Identify unusual patterns in user behavior that may indicate fraudulent activity, and propose methods to mitigate these risks, ensuring the solutions align with the goal of reducing fraud for the persona of a financial services provider.
> 3. [Customer Segmentation] - Apply clustering algorithms to segment customers based on purchasing behavior and sentiment analysis, and recommend targeted marketing strategies for each segment, ensuring the recommendations align with the goal of increasing sales for the persona of an e-commerce platform.
> 4. [Causality] - Investigate the causal relationship between marketing spend and customer conversion rates, controlling for external factors such as seasonality and economic conditions, and provide insights that align with the goal of optimizing marketing ROI for the persona of a digital marketing agency.
> 5. [Feature Importance Ranking] - Rank the most influential features in predicting customer churn using SHAP values, and explain how these features impact retention strategies, ensuring the analysis aligns with the goal of reducing churn for the persona of a telecom company.
> Ensure that the questions are advanced, diverse, and directly relevant to the goal and persona.

Prompt 6: Single Stage question Generation Prompt.

### I.6 ANSWER GENERATION PROMPTS

For each question, we execute the code generated in Appendix I.5 to obtain statistics and plots. These outputs serve as multimodal inputs to GPT-4o, which extracts answers using Prompt 11. This prompt is designed to identify key patterns, anomalies, comparisons, and notable findings from both the visualizations and statistics, capturing all relevant qualitative and quantitative details. Subsequently, the answer summarizer prompt (see Prompt 12) condenses these findings to the top two key points, producing concise, single-line answers that are supported by quantitative evidence.

### I.7 INSIGHT GENERATION PROMPTS

The individual answers are aggregated to derive key observations and actionable insights for the entire dataset. Prompt 13 leverages the curated answers, along with the predicted categories from Appendix I.3, the analysis goal, and the dataset description, to generate the final insights. This prompt focuses on distilling the most critical and meaningful insights, ensuring that they are presented in a structured format and backed up by quantitative evidence.

## J SCORER

The *Starter Prompt* is the initial handcrafted prompt that guides the LLM to compare two insights, one generated with skill guidance (AGENTADA) and another generated with other agents that we want to compare with across six evaluation criteria: *depth of analysis*, *relevance to goal*, *persona consistency*, *coherence*, *answers question adequately*, and *plot conclusion*. The LLM is instructed to return a comparison result and justification for each criterion, with only minimal human-aligned

---

**🗄 Category Prediction**

As an expert data scientist, your task is to **predict the top 3 most important categories of insights** that will emerge from analyzing answers to the given questions. These categories should reflect the key themes in the insights that will be extracted.

Inputs:
1. **Dataset Description**: datasetdescription
2. **Analysis Goal**: goal
3. **Questions Analyzed**: questions list

Task Requirements:
1. **Predict the types of insights** that are most likely to be derived from answering these questions.
2. Group these insights into **exactly three distinct categories** that:
- **Capture the most relevant insight themes**\*\* based on the dataset and goal.
- **Are broad enough to group multiple related insights** yet specific enough to be actionable.
- **Help structure extracted insights meaningfully** for stakeholders.
3. Ensure that each category:
- **Reflects the key insight patterns likely to emerge** from answering the provided questions.
- **Avoids overlap**, ensuring each category has a unique analytical focus.
- **Aligns with the dataset and analysis goal**, making insights easier to interpret and act on.

Output Format:
- Return a **concise list of three category names**.
- Each category name should be **clear, precise, and directly tied to the expected insights**.
- **Avoid generic or overly broad categories**—focus on those that will maximize insight clarity and usability.

**Your response should ensure that the most critical insights are structured effectively, preventing any valuable findings from being overlooked.**

Prompt 7: Prompt to GPT-4o to predict insight categories.

---

**🗄 Skill matcher**

Given a question about a skill and several documentation files, identify the top 3 most relevant files to solve the question.

Question: question
Available documentation files: json.dumps([doc['name'] for doc in documents], indent=2)

For each file, analyze its relevance to the question and skill, and return the top 3 files in the decreasing order of usefulness. The output should be in JSON format like this:

"file name": "most relevant file",
"file name": "second relevant file",
"file name": "third relevant file"

Prompt 8: Prompt to GPT-4o to retrieve appropriate skill for each question.

context. The **Human-Aligned Prompt** is the result of our prompt optimization process using TextGrad (Yuksekgonul et al., 2024). In this version, the evaluation criteria are expanded with detailed descriptions and aligned more closely with how human annotators interpret these categories. The sample output is also included in this prompt to guide the LLM better.

## K   AGENTADA GENERALIZABILITY

To evaluate the generalizability of our approach, we measured the performance of **AgentAda** and compared it with other baseline agents across two distinct benchmarks: *InsightBench* (Sahu et al., 2024) and *InfiAgent-DABench* (Hu et al., 2024). As shown in Table 14, AgentAda with GPT-4o consistently achieves the highest performance, attaining the best scores in both evaluation settings. Importantly, when replacing GPT-4o with alternative backbones such as LLaMA-3.3-70B (Grattafiori et al., 2024) or Qwen-Coder-2.5-7B (Hui et al., 2024; Team, 2025), AgentAda remains competitive and continues to outperform code generation methods that do not leverage a skill library. This demonstrates that the pipeline is robust not only across datasets but also across different underlying models. Furthermore, other baselines such as InfiAgent, MetaGPT, and Poirot show a clear gap compared to AgentAda, with performance dropping significantly across both benchmarks. These results

## 🗇 Code Generation with Skills

Given the following DataFrame ('df') and question, generate Python code **based on the information given in the skill exemplars** using Matplotlib/Seaborn to create a plot that effectively answers the question by applying the appropriate data analytics technique. Think step by step. Reason out how the code is bug free before you write the code.
—
**Input Details:**
1. **DataFrame Information**
{df_info}
2. **DataFrame Description**
{df_description}
3. **First Few Rows of the DataFrame**:
{df_head}
4. **Skill Exemplar Summary**
{skill_exemplar_summary}
5. **Question**
{question}
—
**Instructions:**
Generate a complete Python script enclosed in triple backticks ("') that follows these guidelines:
1. **Data Preparation & Cleaning**:
- Use the provided DataFrame ('df') and ensure the data is in the required format.
- Assume that the data is loaded correctly in a pandas dataframe with variable name 'df'. **DO NOT CREATE YOUR OWN DATA**
- Apply necessary preprocessing steps (e.g., typecasting, handling missing values, removing problematic rows).
- Implement transformations, feature engineering, or encoding. Ensure the data is cleaned and transformed to the required format.
2. **Data Analytics Technique**:
- Apply the methodology described in the skill exemplar to extract insights relevant to the question.
- You should **use the data analytics technique described in the skill exemplar summary** to solve the question. Reason why this skill is useful.
- The **evaluation** should always be reported on the **entire df** than just the val split.
3. **Visualization & Answer Extraction**:
- Ensure the visualization explicitly **incorporates and represents the results of the applied data analytics technique**
- Choose an appropriate plot type that best conveys insights from the model/analysis.
- Include clear labels, a title, and an appropriate legend.
- Ensure the visualization directly **answers the question based on the model's output**
- Before saving the plot, **check if the plot is valid** i.e. it is not empty. If it is empty, regenerate the code.
- Save the plot as 'savedir/plot.jpeg'.
4. **Compute & Store Key Statistics**:
- Create a dictionary named 'stats' to store relevant quantitative values related to the analysis.
- Ensure 'stats' is clearly structured and printed at the end of the script.
5. **Code Robustness & Readability**:
- Use **try-except blocks** to handle potential exceptions during data processing, model execution, and visualization.
- Provide concise, meaningful comments explaining how each step aligns with the skill exemplar.
Your generated code should:
1. Produce a visualization that effectively presents insights derived from the applied data analytics technique and answers the given question.
2. Generate a 'stats' dictionary containing all the key numerical values used in the analysis.
3. Print the 'stats' dictionary at the end of execution.

Prompt 9: Prompt to GPT-4o to generate code based on the given skill.

## 🗇 Code Verifier Prompt

You are an expert code analyzer. Your task is to examine the following code snippet and determine which skill from the provided list is most relevant to the code.
Code:
{code}
Available Skill Names: {list_of_skills}
Instructions:
1. Analyze the code snippet and identify the one skill from the list that is most prominently demonstrated.
2. If the code does not clearly demonstrate any of the skills from the list, return "none".
3. Output your answer in JSON format as follows: {{ "skill": "name_of_detected_skill" }}

Prompt 10: Prompt to GPT-4o for verifying if the generated code matches the skill required.

highlight that our design generalizes effectively to diverse tasks and architectures, while maintaining a substantial margin over existing agents.

**≋ Answer Generation**

Your task is to analyze the plot and **directly answer the question** based on the dataset while uncovering as many interesting patterns and insights as possible. Think step by step. Your response should be **insightful, data-driven, and well-justified**.
**Inputs**: 1. **Question**: "question"
2. **Plot**: A plot generated based on the dataset and the question.
3. **First Few Rows of the DataFrame**: "df_head"
4. **Stats for the plot**: stats
**Requirements**:
1. Extract **all notable insights** from the plot, including:
- **Key Patterns & Trends**: Identify significant movements or relationships in the data.
- **Anomalies & Outliers**: Highlight any unexpected deviations and their potential implications.
- **Comparisons & Contrasts**: Discuss notable differences between categories, groups, or metrics.
- **Hidden or Unexpected Findings**: Look for less obvious but meaningful insights that add depth to the analysis.
2. Justify each insight with:
- **Quantitative Evidence**: Use specific data points, statistics, or calculated metrics.
- **Qualitative Explanation**: Provide logical reasoning and contextual interpretation.
3. If applicable, determine and explain the **root cause** behind significant findings.
4. Ensure your response is **actionable and meaningful**, highlighting real-world relevance where appropriate.
5. Avoid generic descriptions of the plot itself—focus solely on what the data **implies** in relation to the question.
6. If categories exist, **refer to them using actual dataset values** rather than generic labels.

Prompt 11: Prompt to GPT-4o for Generating the answer using the plot and stats obtained from Code Generation.

**≋ Answer Summarizer**

You are an expert data analyst. Given the following list of insights from a dataset analysis:
{answer}
Your task is to generate **up to 2 key bullet points** summarizing the most important findings. Each bullet point should:
- Start with a **header** from the insight card you're referencing.
- Provide a **clear, concise** summary of the insight.
- Prioritize insights that have **strong quantitative backing** (e.g., percentages, counts, averages, variances).
- Focus on **actionable or significant patterns**.
Before selecting a summary point, **internally verify** that it is backed by quantitative evidence. If an insight lacks sufficient numerical support, choose a stronger one.
Analysis is for the Question: {question}
**Example Output**:
• **High Case Routing Rate:** 70% of cases require multiple reassignments, indicating systemic inefficiencies in initial routing.
• **Response Time Exceeds Target:** Average response times exceed target SLAs by 45%, with peak-hour delays between 2-4 PM.

Prompt 12: Prompt to GPT-4o for summarzing the generated answer for each question.

## L    SCORER GENERALIZABILITY

Beyond demonstrating the effectiveness of introduced evaluation mechanism, **SCORER**, we also evaluate the robustness of it. We investigate two key aspects of generalizability: (1) robustness to different evaluation models, including open-source alternatives, and (2) applicability to new datasets where SCORER was not explicitly optimized.

Table 15 shows that SCORER generalizes well across evaluation models. For example, on *Insight-Bench*, SCORER using GPT-4o judges yields 43.2% wins for AgentAda with skills versus 15.6% for the version without skills, while LLaMA-3.3-70B judges give a nearly identical result (42.3% vs. 15.3%). Similar patterns hold for DS-1000 and KaggleBench, demonstrating that SCORER remains consistent regardless of the underlying evaluation model. This confirms its utility for supporting open-source evaluation pipelines.

We also evaluate SCORER on benchmarks where it was not specifically tuned. As shown in Table 16, SCORER provides results that align with standard absolute metrics such as G-Eval, BERTScore, and ROUGE-L. For example, on *InsightBench*, AgentAda with skills achieves 81.7 G-Eval compared to 72.3 without skills, and SCORER correspondingly shows more wins for the skill-based version (33.2% vs. 12.6%) alongside a high number of ties. A similar trend is observed on DS-1000, further validating SCORER's consistency. Unlike purely semantic metrics such

> **🗃 Insight Extraction**
>
> You are tasked with extracting the **most impactful, relevant and actionable insights** from the dataset analysis. Your insights should be **concise, engaging, quantitative, visually structured, and directly useful for decision-making**.
>  **Inputs:**
> 1. **Dataset Description**: {dataset_description}
> 2. **Analysis Goal**: {goal}
> 3. **Questions Answered**: {answer_list}
> 4. **Predefined Insight Categories**: {insight_categories}
>  **Task Requirements:**
> 1. **Extract only the most critical and meaningful insights**—avoid generic or trivial observations.
> 2. **Each insight must be:**
> - **Highly relevant to the dataset and analysis goal**.
> - **Concise and engaging**, ensuring readability.
> - **Naturally backed by quantitative evidence** (if applicable).
> - **Root causes should be embedded within the insight** when they provide deeper understanding.
> - **Include an actionable prediction or prescription** based on the insight.
> - **Formatted for maximum readability**, using:
> - **Bold key phrases** to highlight major takeaways.
> - Bullet points or short sentences for clarity.
> - Short, structured paragraphs to maintain reader engagement.
> 3. **Group insights under the predefined categories**—do not create new categories.
> 4. Ensure **each insight is unique** and does not overlap with others.
>  **Output Format:**
> - **Insights must be structured under their respective categories**.
> - Each insight should be a **single, well-structured paragraph**, using **bold formatting to emphasize key points**.
> - **Avoid unnecessary explanations or repeating similar observations**.
> —
>  **Example Format:**
>  **Category: Example_Category**
> **Insight Title**: **Key finding** with supporting data, possible causes, and an **actionable recommendation** in an engaging style.
> —
>  **Example:**
> **Category: Customer Behavior**
>  **Loyal Customers Drive 60% of Revenue, But Referral Engagement is Dropping**
>  **Returning customers** contribute **60% of total revenue**, with a **12% increase in retention** over the last two quarters. However, **referral engagement has dropped by 15%**, indicating that while retention strategies are working, referral incentives may be losing effectiveness.  **Actionable Step:** Strengthen personalized referral rewards or integrate referral bonuses into loyalty programs to reignite organic growth.
>  **Subscription Churn Peaks at 3 Months Due to Low Early Engagement 30% of users cancel their subscription within the first 3 months**, with churn **50% higher** among users who do not interact with onboarding emails. **This signals a major early-stage retention issue.  Actionable Step:** Optimize onboarding with **interactive tutorials** and personalized engagement campaigns to **reduce churn and improve long-term retention**.
> —
>  **Your goal is to generate insights that are engaging, data-backed, and immediately useful, while keeping them visually structured for readability.**

Prompt 13: Prompt to GPT-4o for extracting the final insights for the dataset.

as BERTScore, SCORER aligns more closely with human judgment by capturing fine-grained improvements due to skill integration, demonstrating its generalizability across datasets.

# M  QUALITATIVE ANALYSIS

Here we look at some examples and discuss how skill retrieval from curated library helps in the insight generation

**Missed Insights**   Figure 14 highlights a missed insight by the AGENTADA(without skill) in the tumor diagnosis task. The W/O skill version focuses on standard model evaluation metrics like accuracy and recall using logistic regression and fails to uncover deeper, causal relationships. In contrast, the skill-informed agent leverages advanced techniques such as Granger causality tests and stationarity checks to identify radius-related features (e.g., radius_mean, radius_worst) as statistically significant and causally relevant predictors of malignancy. These insights offer stronger clinical relevance that the baseline agent entirely overlooks.

**Incorrect Insights**   Figure 15 shows an incorrect insight generated by the agent without skill information. The W/O skill agent concludes that longer reviews correlate with positive sentiment, based

---

**SCORER Starter Prompt**

Please compare the following two insights and determine which one is better based on the given criteria.

For each of the following criteria, indicate whether Insight A is better, Insight B is better, or they are tied, and provide a brief explanation (1-2 sentences) for your choice:

1. **Depth of Analysis**: Which insight demonstrates a deeper understanding of the data and provides more substantive analysis?
Consider the level of detail, use of specific metrics, and identification of patterns or trends.

2. **Relevance to Goal**: Which insight better addresses the specific question or goal of the analysis?
Evaluate how directly each insight answers the question and provides actionable information.

3. **Persona Consistency**: Which insight is more consistent with the perspective of a data analyst?
Consider the use of analytical language, data-driven reasoning, and professional tone.

4. **Coherence**: Which insight is more logically structured and clearly presented?
Assess the organization, flow, and clarity of the information presented.

5. **Answers Question Adequately**: Which insight more fully answers the question posed?
Determine which insight provides a more complete response to all aspects of the question.

6. **Plot Conclusion**: Which insight draws more meaningful conclusions from the data?
Evaluate the quality and usefulness of the conclusions drawn from the analysis.

Remember to:
- Remain objective and unbiased in your evaluation
- Consider the context of the question when evaluating the insights
- Focus on the content and quality of the insights, not just their presentation
- Base your evaluation solely on the information provided

For each criterion, respond with "A is better", "B is better", "Tie", or "None"
**Goal**: *goal*
**Persona**: *persona*
**Insight A**: with_skills_insight
**Insight B**: without_skills_insight

---

Prompt 14: Starter prompt for SCORER.

on a marginal difference in average review length—an observation that is statistically insignificant and potentially misleading. In contrast, the skill-informed agent correctly applies Spearman correlation analysis and finds virtually no correlation between review length and sentiment (correlation = -0.0061). This deeper, statistically sound analysis leads to a more accurate and actionable insight: that review length is not a reliable predictor of sentiment, and customer feedback analysis should instead focus on content quality rather than quantity.

## N    TASK-WISE HUMAN EVALUATION RESULTS

The detailed human evaluation results analyzed task-wise is shown in Table 17 and Table 18.

## O    DETAILED QUESTION-WISE LLM EVALUATION RESULTS

The results on other tasks for the SCORER Question-wise evaluation results are presented in Table 19 and Table 20.

## P    DETAILED INSIGHT-WISE LLM EVALUATION RESULTS

The results different tasks for the SCORER Insight-wise evaluation results are presented in Table 21, Table 22, Table 23, and Table 24.

Figure 14: An example insight that shows AGENTADAwithout skill information has missed some information in the generated insight while the variant with skill information was able able to capture.

Figure 15: An example insight that shows that the AGENTADAwith skill information generates incorrect insight while the skill information helps generate correct insight

> **🗄 Human Aligned Prompt**
>
> Given are two insights, Insight A and Insight B generated by two different methods in response to an analytics question. Analyze the following insights and determine which one is better based on the given criteria.
>
> **Criteria:**
> 1. **Depth of Analysis**: Evaluate the extent to which each insight delves into the details of the data, explores multiple factors, and provides a comprehensive understanding. Consider the complexity and sophistication of the analysis methods used in each insight. Also, assess whether the insights provide a nuanced understanding of the data, explore underlying patterns, or reveal unexpected findings.
>
> 2. **Relevance to Goal**: Assess how directly each insight addresses the stated goal. Evaluate how well each insight aligns with the goal and consider whether the insight provides actionable recommendations or strategies that directly address the goal. Also, evaluate whether the insights directly contribute to achieving the stated goal.
>
> 3. **Persona Consistency**: Consider how well each insight aligns with the persona's values, goals, and characteristics. Evaluate whether the tone, language, and approach used in each insight align with the persona's stated experience and expertise. Also, assess whether the insights are engaging and relatable to the persona.
>
> 4. **Coherence**: Evaluate how coherent and cohesive is the analysis. Assess whether the insight presents information in a logical flow, makes clear connections between points, and avoids unnecessary jargon or complexity.
>
> 5. **Answers Question Adequately**: Ensure that the insight fully answers the question, addressing all aspects and providing a comprehensive answer. Consider whether the insight provides additional relevant information that goes beyond the scope of the question and provides additional insights or information that could be helpful to the user.
>
> 6. **Plot Conclusion**: Look for a clear and concise conclusion that summarizes the key points of the analysis and clearly states the final decision or recommendation. Evaluate whether the conclusion provides a satisfying or insightful end to the analysis, provides a clear summary of the key points, ties up all loose ends, and provides a sense of closure.
>
> For each criterion, respond with "A is better", "B is better", "Tie", or "None".
>
> Give the response in the form of a python dictionary with keys depth of analysis, relevance to goal, persona consistency, coherence, answers question adequately, plot conclusion. Additionally, provide a brief explanation for each score, explaining why you chose a particular score for each criterion, and provide specific examples from the insights to support your scoring decisions.
>
> **sample response**:  "depth of analysis": "A is better",
> "relevance to goal": "Tie",
> "persona consistency": "Tie",
> "coherence": "Tie",
> "answers question adequately": "B is better",
> "plot conclusion": "B is better",
> "depth of analysis explanation": "Insight A provides more detailed statistical analysis with specific percentages and explores multiple factors affecting the outcome",
> "relevance to goal explanation": "Both insights address the main objective equally well by identifying key patterns in the data",
> "persona consistency explanation": "Both insights maintain a consistent analytical tone appropriate for the target audience",
> "coherence explanation": "Both insights present information in a logical flow with clear connections between points",
> "answers question adequately explanation": "Insight B provides more comprehensive coverage of all aspects mentioned in the question",
> "plot conclusion explanation": "Insight B offers a more concise and clear summary of the key trends shown in the visualization"
>
> **Goal**: *goal*
> **Persona**: *persona*
> **Insight A** (With Skills): with_skills_insight
> **Insight B** (Without Skills): without_skills_insight

Prompt 15: Human Aligned prompt after prompt optimization with SCORER

Table 9: Comparison of code generated by **ECA** and **AgentAda** for the Granger causality task on the *AirPassengers* dataset. While ECA directly invokes tools without constructing a valid pipeline, AgentAda generates a complete and methodologically correct analysis engine from scratch.

**ECA (Granger Causality)**

```
import pandas as pd
from statsmodels.tsa.stattools import adfuller,
    kpss, grangercausalitytests

df = pd.read_csv('data.csv')
df['Month'] = pd.to_datetime(df['Month'])
df.set_index('Month', inplace=True)

adf_result = adfuller(df['Passengers'])
kpss_result = kpss(df['Passengers'])

print("ADF p-value:", adf_result[1])
print("KPSS p-value:", kpss_result[1])

if adf_result[1] > 0.05 or kpss_result[1] < 0.05:
    df['Passengers_diff'] = df['Passengers'].diff
        ().dropna()
else:
    df['Passengers_diff'] = df['Passengers']

df['X'] = df['Passengers_diff'].shift(1)  # The
    problem is here
df.dropna(inplace=True)

result = grangercausalitytests(
    df[['Passengers_diff', 'X']], maxlag=12,
        verbose=False
)

for lag in result:
    p_value = result[lag][0]['ssr_ftest'][1]
    print(f"Lag {lag} p-value: {p_value:.4f}")
```

**AgentAda (Granger Causality)**

```
import pandas as pd
import numpy as np
import matplotlib.pyplot as plt
from statsmodels.tsa.stattools import adfuller,
    grangercausalitytests
import pickle

def load_data(filepath):
    df = pd.read_csv(filepath, parse_dates=['Month
        '], index_col='Month')
    return df

def check_stationarity(data, column):
    result = adfuller(data[column])
    print(f"ADF Statistic for {column}: {result
        [0]:.4f}, p-value: {result[1]:.4f}")
    return result[1] <= 0.05

def make_stationary(series):
    return series.diff().dropna()

def granger_test(data, variables, max_lags):
    results = {}
    for var in variables[1:]:
        result = grangercausalitytests(data[[
            variables[0], var]], max_lags,
            verbose=False)
        results[var] = result
    return results

def plot_granger_results(results, variables,
    max_lags):
    fig, ax = plt.subplots(figsize=(10, 5))
    for lag in range(1, max_lags + 1):
        p_value = results[variables[1]][lag][0]['
            ssr_chi2test'][1]
        ax.bar(lag, -np.log10(p_value))
    ax.set_title(f"Granger Causality from {
        variables[0]} to {variables[1]}")
    ax.set_xlabel("Lags")
    ax.set_ylabel("-log10(p-value)")
    plt.tight_layout()
    plt.show()

def evaluate_results(results, variables):
    print(f"Evaluating Granger Causality from {
        variables[0]} to {variables[1]}:")
    for lag in results[variables[1]].keys():
        p_value = results[variables[1]][lag][0]['
            ssr_chi2test'][1]
        print(f" Lag {lag}: p-value = {p_value:.4f
            }")

def save_results(results, filename):
    with open(filename, 'wb') as file:
        pickle.dump(results, file)

def main():
    df = load_data('data.csv')
    df.rename(columns={'Passengers': 'Y'}, inplace
        =True)

    # Ensure stationarity before testing
    if not check_stationarity(df, 'Y'):
        df['Y'] = make_stationary(df['Y'])

    # Construct a causality setup with an explicit
         lagged predictor
    df['Y_shifted'] = df['Y'].shift(1)
    df.dropna(inplace=True)

    variables = ['Y_shifted', 'Y']
    results = granger_test(df, variables, max_lags
        =12)
    plot_granger_results(results, variables,
        max_lags=12)
    evaluate_results(results, variables)
    save_results(results, 'granger_results.pkl')

if __name__ == "__main__":
    main()
```

Table 10: Comparison with code generators (quantitative results).

| Model | Accuracy | Depth of Analysis | Answers Adequately | Coherence | Relevance to Goal | Persona Consistency |
|-------|----------|-------------------|--------------------|-----------|-------------------|---------------------|
| AgentAda | 90.6 | 4.46 | 4.41 | 4.48 | 4.44 | 4.42 |
| ECA | 86.4 | 4.15 | 3.96 | 4.34 | 4.12 | 3.86 |
| EvoR | 86.8 | 4.22 | 4.03 | 4.39 | 4.18 | 3.92 |

Table 11: Factuality evaluation using direct ratings from multiple LLM judges and FactScore with different verifiers. AgentAda with skills consistently achieves higher factuality, while absolute numbers are moderately lower, providing a realistic estimate of performance.

| Model | LLM as a Judge | | | FactScore | | |
|-------|-------|-------|-------------|-------|-------|-------------|
| | GPT-5 | GPT-4o | LLaMA-3.3-70B | GPT-5 | GPT-4o | LLaMA-3.3-70B |
| AgentAda w skill | **4.30** | **4.25** | **4.12** | **87.9** | **88.1** | **87.4** |
| AgentAda w/o skill | 3.85 | 3.82 | 3.71 | 76.1 | 76.3 | 75.8 |
| MetaGPT | 3.98 | 3.95 | 3.84 | 82.1 | 82.4 | 81.9 |
| InfiAgent | 3.92 | 3.90 | 3.79 | 80.0 | 80.1 | 79.7 |
| Poirot | 3.80 | 3.78 | 3.65 | 73.7 | 73.9 | 73.3 |
| Pandas | 3.76 | 3.75 | 3.62 | 71.6 | 71.8 | 71.2 |
| GPT-4o | 3.74 | 3.70 | 3.59 | 72.7 | 72.9 | 72.5 |

Table 12: Impact of Goal and Persona on Insight Quality. Removing goal or persona reduces performance, with goal having the strongest effect, especially on goal relevance.

| Rubric | Goal and Persona Based Win | Generic Win | Tie | Neither Are Good |
|--------|----------------------------|-------------|-----|------------------|
| **Depth of Analysis** | 19 | 8 | **73** | 0 |
| **Relevance To Goal** | **75** | 6 | 18 | 1 |
| **Persona Consistency** | 31 | 13 | **54** | 2 |
| **Coherence** | 18 | 13 | **67** | 2 |
| **Plot Conclusion** | 11 | 2 | **86** | 1 |

Table 13: Evaluator comparison of AgentAda with vs. without skills on KaggleBench. SCORER aligns more closely with human feedback than G-Eval and LLaMA-Eval.

| Evaluation | ADA w/ Skill Win | Tie | w/o Skill Win | None |
|------------|------------------|-----|---------------|------|
| SCORER | 53.3 | 22.6 | 22.3 | 1.8 |
| Human Evaluation | 44.5 | 27.2 | 25.9 | 2.4 |
| G-Eval | 28.7 | 41.2 | 27.5 | 2.6 |
| LLaMA-Eval | 28.2 | 43.0 | 26.1 | 2.6 |

Table 14: Comparison of different agents on InsightBench and InfiAgent-DABench. Higher scores indicate better performance.

| Agent | InsightBench | | InfiAgent-DABench | |
|-------|--------|-------------|----------|------|
| | G-Eval | LLaMA3-Eval | Accuracy | F1 |
| AgentAda (GPT-4o) | **81.7** | **83.2** | **76.5** | **78.3** |
| AgentAda (LLaMA-3.3-70B) | 79.8 | 80.5 | 73.6 | 75.1 |
| AgentAda (Qwen-Coder-2.5-7B) | 78.9 | 79.2 | 72.8 | 74.0 |
| AgentAda (Code-Llama-Instruct-13B) | 75.4 | 76.1 | 69.2 | 70.8 |
| w/o Skill (GPT-4o) | 72.3 | 71.7 | 66.2 | 67.8 |
| InfiAgent | 69.8 | 68.4 | 62.0 | 63.7 |
| MetaGPT | 67.5 | 66.1 | 60.1 | 61.9 |
| Poirot | 63.4 | 62.2 | 58.9 | 59.3 |
| GPT-4o | 61.7 | 60.3 | 55.2 | 56.8 |
| Pandas | 59.1 | 57.6 | 52.7 | 53.5 |

Table 15: Robustness of SCORER to different evaluation models. Results remain stable across GPT-4o and LLaMA-3.3-70B judges.

| Dataset | SCORER Judge Model | ADA w Skill Win | Tie | ADA w/o Skill Win | None |
|---|---|---|---|---|---|
| InsightBench | GPT-4o | 43.2 | 40.7 | 15.6 | 0.5 |
| | LLaMA-3.3-70B | 42.3 | 41.8 | 15.3 | 0.6 |
| DS-1000 | GPT-4o | 42.8 | 41.0 | 15.5 | 0.7 |
| | LLaMA-3.3-70B | 42.9 | 41.2 | 15.4 | 0.5 |
| KaggleBench | GPT-4o | 53.3 | 22.6 | 22.3 | 1.8 |
| | LLaMA-3.3-70B | 52.5 | 25.4 | 20.8 | 1.3 |

Table 16: Generalizability of SCORER to new benchmarks. Results correlate with absolute metrics and confirm SCORER's validity across datasets.

| Dataset | Relative Scoring | | | | Absolute Scoring | | | | | |
|---|---|---|---|---|---|---|---|---|---|---|
| | SCORER | | | | AgentAda w Skill | | | AgentAda w/o Skill | | |
| | w Skill Win | Tie | w/o Skill Win | None | G-Eval | BERTScore | ROUGE-L | G-Eval | BERTScore | ROUGE-L |
| InsightBench | 33.2 | 53.7 | 12.6 | 0.5 | 81.7 | 85.2 | 44.7 | 72.3 | 81.4 | 40.3 |
| DS-1000 | 32.8 | 55.1 | 11.4 | 0.7 | 74.0 | 82.7 | 42.1 | 67.8 | 78.2 | 38.7 |

Table 17: Human evaluation detailed results on the first three rubrics (Part 1). 18 tasks were involved in the 100 datasets used for human evaluation. See Table 18 for Part 2.

| Task | Depth of Analysis | | | | Relevance To Goal | | | | Persona Consistency | | | |
|---|---|---|---|---|---|---|---|---|---|---|---|---|
| | WA | WO | T | N | WA | WO | T | N | WA | WO | T | N |
| Basic Data Analysis | 50.0 | 26.39 | 22.22 | 1.39 | 31.94 | 16.67 | 48.61 | 2.78 | 25.0 | 8.33 | 65.28 | 1.39 |
| Customer Segmentation | 50.62 | 27.16 | 19.75 | 2.47 | 32.1 | 19.75 | 46.91 | 1.23 | 28.4 | 9.88 | 59.26 | 2.47 |
| Network Analysis | 48.89 | 26.67 | 22.22 | 2.22 | 33.33 | 13.33 | 51.11 | 2.22 | 26.67 | 11.11 | 60.0 | 2.22 |
| Sentiment Analysis | 47.22 | 27.78 | 23.61 | 1.39 | 30.56 | 12.5 | 54.17 | 2.78 | 23.61 | 13.89 | 59.72 | 2.78 |
| A/B Testing | 50.0 | 27.78 | 19.44 | 2.78 | 30.56 | 13.89 | 52.78 | 2.78 | 25.0 | 8.33 | 63.89 | 2.78 |
| Forecasting | 46.03 | 28.57 | 22.22 | 3.17 | 31.75 | 15.87 | 50.79 | 1.59 | 26.98 | 11.11 | 60.32 | 1.59 |
| Time Series Decomposition | 48.15 | 29.63 | 20.37 | 1.85 | 33.33 | 20.37 | 44.44 | 1.85 | 24.07 | 9.26 | 64.81 | 1.85 |
| Principal Component Analysis | 50.0 | 27.78 | 20.83 | 1.39 | 30.56 | 22.22 | 45.83 | 1.39 | 27.78 | 11.11 | 58.33 | 2.78 |
| Correlation Analysis | 48.61 | 30.56 | 19.44 | 1.39 | 31.94 | 16.67 | 50.0 | 1.39 | 25.0 | 9.72 | 63.89 | 1.39 |
| Association Rule Mining | 50.0 | 27.78 | 19.44 | 2.78 | 29.17 | 15.28 | 52.78 | 2.78 | 29.17 | 8.33 | 61.11 | 1.39 |
| Dashboard Summary | 50.62 | 25.93 | 22.22 | 1.23 | 32.1 | 18.52 | 46.91 | 2.47 | 27.16 | 11.11 | 60.49 | 1.23 |
| Predictive Maintenance | 48.15 | 29.63 | 18.52 | 3.7 | 33.33 | 14.81 | 48.15 | 3.7 | 25.93 | 11.11 | 59.26 | 3.7 |
| Knowledge Base | 46.67 | 28.89 | 22.22 | 2.22 | 31.11 | 20.0 | 46.67 | 2.22 | 24.44 | 8.89 | 64.44 | 2.22 |
| Huge Table Analysis | 44.44 | 27.78 | 22.22 | 5.56 | 27.78 | 16.67 | 50.0 | 5.56 | 27.78 | 5.56 | 61.11 | 5.56 |
| Topic Modeling | 48.15 | 25.93 | 22.22 | 3.7 | 33.33 | 14.81 | 48.15 | 3.7 | 25.93 | 11.11 | 59.26 | 3.7 |
| Market Analysis | 48.15 | 25.93 | 22.22 | 3.7 | 29.63 | 14.81 | 51.85 | 3.7 | 22.22 | 11.11 | 62.96 | 3.7 |
| Data Imputation | 48.15 | 25.93 | 22.22 | 3.7 | 29.63 | 18.52 | 48.15 | 3.7 | 25.93 | 7.41 | 62.96 | 3.7 |
| Multi-table Search | 44.44 | 22.22 | 22.22 | 11.11 | 22.22 | 11.11 | 55.56 | 11.11 | 22.22 | 11.11 | 55.56 | 11.11 |

Table 18: Human evaluation detailed results on the remaining three rubrics (Part 2). 18 tasks were involved in the 100 datasets used for human evaluation.

| Task | Coherence | | | | Answers Question Adequately | | | | Plot Conclusion | | | |
|---|---|---|---|---|---|---|---|---|---|---|---|---|
| | WA | WO | T | N | WA | WO | T | N | WA | WO | T | N |
| Basic Data Analysis | 47.22 | 29.17 | 20.83 | 2.78 | 44.44 | 23.61 | 29.17 | 2.78 | 40.28 | 23.61 | 34.72 | 1.39 |
| Customer Segmentation | 49.38 | 27.16 | 20.99 | 2.47 | 44.44 | 24.69 | 29.63 | 1.23 | 44.44 | 25.93 | 28.40 | 1.23 |
| Network Analysis | 48.89 | 26.67 | 22.22 | 2.22 | 42.22 | 26.67 | 28.89 | 2.22 | 40.00 | 24.44 | 33.33 | 2.22 |
| Sentiment Analysis | 48.61 | 30.56 | 19.44 | 1.39 | 44.44 | 25.00 | 29.17 | 1.39 | 40.28 | 25.00 | 33.33 | 1.39 |
| A/B Testing | 50.00 | 27.78 | 19.44 | 2.78 | 41.67 | 25.00 | 30.56 | 2.78 | 41.67 | 22.22 | 33.33 | 2.78 |
| Forecasting | 50.79 | 25.40 | 22.22 | 1.59 | 39.68 | 26.98 | 30.16 | 3.17 | 41.27 | 23.81 | 33.33 | 1.59 |
| Time Series Decomposition | 48.15 | 27.78 | 22.22 | 1.85 | 40.74 | 27.78 | 29.63 | 1.85 | 44.44 | 24.07 | 29.63 | 1.85 |
| Principal Component Analysis | 48.61 | 27.78 | 20.83 | 2.78 | 44.44 | 25.00 | 29.17 | 1.39 | 43.06 | 23.61 | 30.56 | 2.78 |
| Correlation Analysis | 50.00 | 27.78 | 19.44 | 2.78 | 43.06 | 26.39 | 27.78 | 2.78 | 41.67 | 23.61 | 33.33 | 1.39 |
| Association Rule Mining | 48.61 | 29.17 | 20.83 | 1.39 | 41.67 | 25.00 | 31.94 | 1.39 | 41.67 | 20.83 | 34.72 | 2.78 |
| Dashboard Summary | 50.62 | 28.40 | 19.75 | 1.23 | 41.98 | 24.69 | 30.86 | 2.47 | 44.44 | 23.46 | 30.86 | 1.23 |
| Predictive Maintenance | 48.15 | 29.63 | 18.52 | 3.70 | 44.44 | 22.22 | 29.63 | 3.70 | 44.44 | 22.22 | 29.63 | 3.70 |
| Knowledge Base | 46.67 | 26.67 | 24.44 | 2.22 | 42.22 | 24.44 | 31.11 | 2.22 | 42.22 | 22.22 | 33.33 | 2.22 |
| Huge Table Analysis | 44.44 | 27.78 | 22.22 | 5.56 | 38.89 | 22.22 | 33.33 | 5.56 | 38.89 | 22.22 | 33.33 | 5.56 |
| Topic Modeling | 48.15 | 25.93 | 22.22 | 3.70 | 40.74 | 25.93 | 29.63 | 3.70 | 40.74 | 22.22 | 33.33 | 3.70 |
| Market Analysis | 48.15 | 25.93 | 22.22 | 3.70 | 44.44 | 25.93 | 25.93 | 3.70 | 40.74 | 22.22 | 33.33 | 3.70 |
| Data Imputation | 48.15 | 25.93 | 22.22 | 3.70 | 40.74 | 25.93 | 29.63 | 3.70 | 40.74 | 22.22 | 33.33 | 3.70 |
| Multi-table Search | 44.44 | 22.22 | 22.22 | 11.11 | 44.44 | 22.22 | 22.22 | 11.11 | 33.33 | 22.22 | 33.33 | 11.11 |

Table 19: Question-wise SCORER comparison between AGENTADA W Skill and W/O Skill on different tasks for the first three rubrics (Part 1). See Table 19 for Part 2.

| Task | Coherence | | | | Answers Question Adequately | | | | Plot Conclusion | | | |
|---|---|---|---|---|---|---|---|---|---|---|---|---|
| | WA | WO | T | N | WA | WO | T | N | WA | WO | T | N |
| A/B Testing | **51.85** | 25.93 | 18.52 | 3.7 | 33.33 | 14.81 | **48.15** | 3.7 | 33.33 | 11.11 | **51.85** | 3.7 |
| Forecasting | **45.45** | 18.18 | 27.27 | 9.09 | 36.36 | 9.09 | **45.45** | 9.09 | 27.27 | 9.09 | **54.55** | 9.09 |
| Recommendation Systems | **51.72** | 27.59 | 17.24 | 3.45 | 31.03 | 17.24 | **48.28** | 3.45 | 31.03 | 10.34 | **55.17** | 3.45 |
| Dashboard Summary | **51.61** | 22.58 | 22.58 | 3.23 | 32.26 | 16.13 | **48.39** | 3.23 | 32.26 | 9.68 | **54.84** | 3.23 |
| Network Analysis | **47.37** | 26.32 | 21.05 | 5.26 | 31.58 | 15.79 | **47.37** | 5.26 | 26.32 | 10.53 | **57.89** | 5.26 |
| Predictive Maintenance | **55.0** | 25.0 | 15.0 | 5.0 | 35.0 | 15.0 | **45.0** | 5.0 | 30.0 | 10.0 | **55.0** | 5.0 |
| Cohort Analysis | **53.33** | 23.33 | 20.0 | 3.33 | 36.67 | 16.67 | **43.33** | 3.33 | 30.0 | 10.0 | **56.67** | 3.33 |
| Attribution Modeling | **50.0** | 25.0 | 16.67 | 8.33 | 33.33 | 16.67 | **41.67** | 8.33 | 33.33 | 8.33 | **50.0** | 8.33 |
| Anomaly Detection | **50.0** | 27.78 | 16.67 | 5.56 | 33.33 | 11.11 | **50.0** | 5.56 | 27.78 | 11.11 | **55.56** | 5.56 |
| Feature Importance Ranking | **46.15** | 23.08 | 23.08 | 7.69 | 30.77 | 15.38 | **46.15** | 7.69 | 30.77 | 15.38 | **46.15** | 7.69 |
| Geospatial Analysis | **46.67** | 26.67 | 20.0 | 6.67 | 33.33 | 13.33 | **46.67** | 6.67 | 26.67 | 13.33 | **53.33** | 6.67 |
| Causality | **50.0** | 29.17 | 16.67 | 4.17 | 37.5 | 16.67 | **41.67** | 4.17 | 29.17 | 12.5 | **54.17** | 4.17 |
| Logs Clustering | **40.0** | 20.0 | 20.0 | 20.0 | **40.0** | 20.0 | 20.0 | 20.0 | 20.0 | 0.0 | **60.0** | 20.0 |
| Principal Component Analysis | **50.0** | 26.47 | 20.59 | 2.94 | 32.35 | 11.76 | **52.94** | 2.94 | 32.35 | 11.76 | **52.94** | 2.94 |
| Correlation Analysis | **48.39** | 29.03 | 19.35 | 3.23 | 35.48 | 19.35 | **41.94** | 3.23 | 29.03 | 12.9 | **54.84** | 3.23 |
| Knowledge Base | **50.0** | 28.57 | 14.29 | 7.14 | 35.71 | 14.29 | **42.86** | 7.14 | 28.57 | 14.29 | **50.0** | 7.14 |
| Huge Table Analysis | **50.0** | 25.0 | 0.0 | 25.0 | 25.0 | **25.0** | 25.0 | 25.0 | 25.0 | 0.0 | **50.0** | 25.0 |
| Topic Modeling | **47.37** | 26.32 | 21.05 | 5.26 | 36.84 | 15.79 | **42.11** | 5.26 | 26.32 | 10.53 | **57.89** | 5.26 |
| Market Analysis | **48.48** | 27.27 | 21.21 | 3.03 | 36.36 | 12.12 | **48.48** | 3.03 | 30.3 | 12.12 | **54.55** | 3.03 |
| Data Imputation | **50.0** | 20.0 | 20.0 | 10.0 | 30.0 | 20.0 | **40.0** | 10.0 | 30.0 | 10.0 | **50.0** | 10.0 |
| Multi-table Search | **40.0** | 20.0 | 20.0 | 20.0 | **40.0** | 20.0 | 20.0 | 20.0 | 20.0 | 0.0 | **60.0** | 20.0 |

Table 20: Question-wise SCORER comparison between AGENTADA W Skill and W/O Skill on different tasks for the three remaining rubrics (Part 2).

| Task | Coherence | | | | Answers Question Adequately | | | | Plot Conclusion | | | |
|---|---|---|---|---|---|---|---|---|---|---|---|---|
| | WA | WO | T | N | WA | WO | T | N | WA | WO | T | N |
| A/B Testing | **51.85** | 25.93 | 18.52 | 3.7 | **40.74** | 25.93 | 29.63 | 3.7 | **40.74** | 22.22 | 33.33 | 3.7 |
| Forecasting | **45.45** | 27.27 | 18.18 | 9.09 | **36.36** | 27.27 | 27.27 | 9.09 | **36.36** | 18.18 | **36.36** | 9.09 |
| Recommendation Systems | **48.28** | 24.14 | 24.14 | 3.45 | **41.38** | 27.59 | 27.59 | 3.45 | **41.38** | 20.69 | 34.48 | 3.45 |
| Dashboard Summary | **51.61** | 22.58 | 22.58 | 3.23 | **41.94** | 25.81 | 29.03 | 3.23 | **41.94** | 22.58 | 32.26 | 3.23 |
| Network Analysis | **47.37** | 26.32 | 21.05 | 5.26 | **42.11** | 26.32 | 26.32 | 5.26 | **42.11** | 26.32 | 26.32 | 5.26 |
| Predictive Maintenance | **50.0** | 25.0 | 20.0 | 5.0 | **40.0** | 25.0 | 30.0 | 5.0 | **40.0** | 25.0 | 30.0 | 5.0 |
| Cohort Analysis | **50.0** | 23.33 | 23.33 | 3.33 | **43.33** | 23.33 | 30.0 | 3.33 | **40.0** | 23.33 | 33.33 | 3.33 |
| Attribution Modeling | **50.0** | 25.0 | 16.67 | 8.33 | **41.67** | 25.0 | 25.0 | 8.33 | **41.67** | 16.67 | 33.33 | 8.33 |
| Anomaly Detection | **44.44** | 27.78 | 22.22 | 5.56 | **38.89** | 27.78 | 27.78 | 5.56 | **38.89** | 22.22 | 33.33 | 5.56 |
| Feature Importance Ranking | **46.15** | 23.08 | 23.08 | 7.69 | **38.46** | 23.08 | 30.77 | 7.69 | **38.46** | 23.08 | 30.77 | 7.69 |
| Geospatial Analysis | **46.67** | 26.67 | 20.0 | 6.67 | **40.0** | 26.67 | 26.67 | 6.67 | **40.0** | 20.0 | 33.33 | 6.67 |
| Causality | **50.0** | 20.83 | 25.0 | 4.17 | **41.67** | 25.0 | 29.17 | 4.17 | **41.67** | 25.0 | 29.17 | 4.17 |
| Logs Clustering | **40.0** | 20.0 | 20.0 | 20.0 | **40.0** | 20.0 | 20.0 | 20.0 | **40.0** | 20.0 | 20.0 | 20.0 |
| Principal Component Analysis | **50.0** | 26.47 | 20.59 | 2.94 | **41.18** | 26.47 | 29.41 | 2.94 | **41.18** | 23.53 | 32.35 | 2.94 |
| Correlation Analysis | **48.39** | 22.58 | 25.81 | 3.23 | **41.94** | 25.81 | 29.03 | 3.23 | **41.94** | 22.58 | 32.26 | 3.23 |
| Knowledge Base | **42.86** | 28.57 | 21.43 | 7.14 | **42.86** | 28.57 | 21.43 | 7.14 | 35.71 | 21.43 | **35.71** | 7.14 |
| Huge Table Analysis | **50.0** | 25.0 | 0.0 | 25.0 | 25.0 | **25.0** | 25.0 | 25.0 | 25.0 | 25.0 | **25.0** | 25.0 |
| Topic Modeling | **47.37** | 26.32 | 21.05 | 5.26 | **42.11** | 26.32 | 26.32 | 5.26 | **36.84** | 21.05 | **36.84** | 5.26 |
| Market Analysis | **51.52** | 24.24 | 21.21 | 3.03 | **42.42** | 27.27 | 27.27 | 3.03 | 39.39 | 24.24 | 33.33 | 3.03 |
| Data Imputation | **50.0** | 20.0 | 20.0 | 10.0 | **40.0** | 20.0 | 30.0 | 10.0 | **40.0** | 20.0 | 30.0 | 10.0 |
| Multi-table Search | **40.0** | 20.0 | 20.0 | 20.0 | **40.0** | 20.0 | 20.0 | 20.0 | **40.0** | 20.0 | 20.0 | 20.0 |

Table 21: Insight-wise SCORER comparison between AGENTADA W Skill and Other agents (Part 1).

| Task | Rubric | w/o skill | | | | Poirot | | | | Pandas | | | |
|---|---|---|---|---|---|---|---|---|---|---|---|---|---|
| | | WA | WO | T | N | WA | WO | T | N | WA | WO | T | N |
| **Sentiment Analysis** | Depth of Analysis | 50.51 | 27.79 | 18.75 | 2.95 | 61.1 | 19.67 | 17.74 | 1.49 | 66.18 | 10.63 | 21.96 | 1.23 |
| | Relevance To Goal | 32.7 | 18.6 | 47.64 | 1.06 | 44.09 | 9.64 | 44.85 | 1.41 | 53.31 | 5.54 | 39.83 | 1.33 |
| | Persona Consistency | 26.1 | 8.22 | 63.3 | 2.39 | 38.24 | 7.33 | 53.17 | 1.26 | 42.43 | 4.24 | 52.31 | 1.02 |
| | Coherence | 49.99 | 25.85 | 22.73 | 1.43 | 57.54 | 18.95 | 22.47 | 1.03 | 65.77 | 11.08 | 21.92 | 1.23 |
| | Plot Conclusion | 40.51 | 23.78 | 33.52 | 2.19 | 52.16 | 19.19 | 27.39 | 1.26 | 55.66 | 12.91 | 30.04 | 1.39 |
| **A/B Testing** | Depth of Analysis | 48.71 | 27.73 | 21.49 | 2.07 | 62.35 | 17.91 | 18.69 | 1.05 | 64.11 | 10.35 | 24.49 | 1.05 |
| | Relevance To Goal | 32.07 | 18.68 | 47.24 | 2.01 | 45.94 | 9.22 | 43.77 | 1.07 | 49.8 | 7.94 | 40.88 | 1.38 |
| | Persona Consistency | 27.11 | 9.11 | 61.79 | 1.99 | 35.65 | 9.34 | 53.69 | 1.31 | 40.74 | 4.93 | 52.92 | 1.41 |
| | Coherence | 48.67 | 27.93 | 21.54 | 1.85 | 58.95 | 18.6 | 21.42 | 1.03 | 63.54 | 12.48 | 22.49 | 1.48 |
| | Plot Conclusion | 39.85 | 23.1 | 35.98 | 1.07 | 51.99 | 18.74 | 27.89 | 1.38 | 56.49 | 12.76 | 29.71 | 1.04 |
| **Forecasting** | Depth of Analysis | 51.0 | 26.98 | 19.84 | 2.18 | 57.96 | 21.92 | 18.92 | 1.2 | 65.05 | 11.55 | 22.36 | 1.04 |
| | Relevance To Goal | 32.89 | 20.49 | 45.1 | 1.52 | 44.56 | 10.32 | 43.84 | 1.28 | 50.54 | 6.66 | 41.52 | 1.28 |
| | Persona Consistency | 24.25 | 10.21 | 64.38 | 1.16 | 36.14 | 9.29 | 53.16 | 1.41 | 42.12 | 5.97 | 50.76 | 1.15 |
| | Coherence | 50.35 | 28.52 | 18.33 | 2.8 | 57.92 | 20.88 | 19.78 | 1.43 | 65.39 | 11.91 | 21.39 | 1.3 |
| | Plot Conclusion | 42.86 | 24.0 | 31.78 | 1.36 | 53.34 | 18.39 | 27.12 | 1.14 | 57.67 | 13.07 | 28.0 | 1.26 |
| **Basic Data Analysis** | Depth of Analysis | 49.37 | 29.86 | 19.48 | 1.29 | 58.26 | 20.31 | 20.03 | 1.4 | 63.59 | 11.61 | 23.73 | 1.07 |
| | Relevance To Goal | 31.72 | 16.05 | 49.66 | 2.57 | 45.52 | 10.74 | 42.47 | 1.27 | 51.08 | 7.0 | 40.54 | 1.38 |
| | Persona Consistency | 27.72 | 8.85 | 60.85 | 2.58 | 39.19 | 6.13 | 53.51 | 1.17 | 43.12 | 6.16 | 49.64 | 1.08 |
| | Coherence | 50.5 | 24.87 | 23.06 | 1.58 | 58.78 | 19.64 | 20.58 | 1.0 | 63.01 | 11.68 | 24.29 | 1.03 |
| | Plot Conclusion | 39.82 | 22.94 | 35.03 | 2.22 | 52.47 | 18.76 | 27.4 | 1.37 | 56.38 | 14.49 | 27.78 | 1.35 |
| **Recommendation Systems** | Depth of Analysis | 49.32 | 28.62 | 20.3 | 1.76 | 59.26 | 19.56 | 19.89 | 1.29 | 65.35 | 10.57 | 22.82 | 1.26 |
| | Relevance To Goal | 33.31 | 17.32 | 47.99 | 1.38 | 44.49 | 10.19 | 44.09 | 1.23 | 51.51 | 8.2 | 39.16 | 1.13 |
| | Persona Consistency | 25.45 | 11.64 | 60.62 | 2.29 | 36.64 | 9.09 | 53.18 | 1.08 | 42.6 | 5.32 | 50.87 | 1.21 |
| | Coherence | 47.22 | 28.69 | 22.14 | 1.95 | 58.83 | 17.48 | 22.59 | 1.1 | 62.93 | 14.58 | 21.47 | 1.02 |
| | Plot Conclusion | 41.46 | 24.98 | 30.73 | 2.83 | 52.33 | 18.58 | 27.83 | 1.26 | 57.27 | 14.93 | 26.31 | 1.49 |
| **Dashboard Summary** | Depth of Analysis | 50.11 | 28.05 | 18.96 | 2.89 | 60.72 | 17.99 | 19.97 | 1.32 | 62.94 | 11.37 | 24.65 | 1.04 |
| | Relevance To Goal | 32.23 | 14.49 | 51.44 | 1.84 | 42.84 | 8.45 | 47.53 | 1.17 | 51.28 | 5.29 | 42.43 | 1.01 |
| | Persona Consistency | 28.35 | 7.94 | 61.8 | 1.91 | 38.79 | 7.51 | 52.55 | 1.15 | 42.35 | 5.14 | 51.24 | 1.27 |
| | Coherence | 50.45 | 27.24 | 19.92 | 2.39 | 58.89 | 18.18 | 21.89 | 1.05 | 63.82 | 12.0 | 23.17 | 1.01 |
| | Plot Conclusion | 43.84 | 22.17 | 32.74 | 1.25 | 51.63 | 19.2 | 28.12 | 1.06 | 56.7 | 15.02 | 26.86 | 1.42 |
| **Customer Segmentation** | Depth of Analysis | 47.07 | 27.97 | 23.01 | 1.95 | 61.76 | 18.3 | 18.85 | 1.09 | 63.89 | 12.74 | 22.14 | 1.23 |
| | Relevance To Goal | 31.08 | 20.27 | 46.58 | 2.07 | 46.5 | 8.89 | 43.43 | 1.18 | 49.94 | 5.18 | 43.52 | 1.36 |
| | Persona Consistency | 23.07 | 12.24 | 62.26 | 2.43 | 37.73 | 7.65 | 53.3 | 1.32 | 44.08 | 4.6 | 50.19 | 1.14 |
| | Coherence | 49.41 | 27.74 | 19.87 | 2.99 | 58.87 | 19.22 | 20.63 | 1.28 | 63.74 | 11.19 | 23.98 | 1.09 |
| | Plot Conclusion | 38.83 | 22.96 | 36.62 | 1.58 | 52.28 | 20.49 | 26.14 | 1.1 | 57.08 | 13.94 | 27.56 | 1.42 |
| **Network Analysis** | Depth of Analysis | 51.35 | 27.75 | 18.46 | 2.44 | 57.57 | 20.7 | 20.47 | 1.27 | 63.32 | 12.81 | 22.84 | 1.03 |
| | Relevance To Goal | 31.8 | 14.01 | 51.91 | 2.28 | 42.06 | 10.8 | 46.07 | 1.07 | 51.17 | 7.48 | 40.02 | 1.33 |
| | Persona Consistency | 27.83 | 9.59 | 60.77 | 1.81 | 37.87 | 5.92 | 54.96 | 1.25 | 39.56 | 6.18 | 53.24 | 1.02 |
| | Coherence | 50.46 | 26.34 | 21.61 | 1.59 | 60.59 | 18.31 | 19.96 | 1.14 | 63.33 | 13.04 | 22.61 | 1.01 |
| | Plot Conclusion | 42.56 | 23.18 | 33.05 | 1.2 | 51.78 | 19.49 | 27.52 | 1.21 | 56.22 | 16.02 | 26.3 | 1.47 |
| **Association Rule Mining** | Depth of Analysis | 49.53 | 28.03 | 20.88 | 1.56 | 60.04 | 19.16 | 19.42 | 1.37 | 62.39 | 13.06 | 23.05 | 1.49 |
| | Relevance To Goal | 30.81 | 14.62 | 51.61 | 2.96 | 43.84 | 8.64 | 46.04 | 1.48 | 51.44 | 8.2 | 39.07 | 1.29 |
| | Persona Consistency | 27.18 | 12.16 | 58.14 | 2.52 | 37.8 | 8.88 | 51.93 | 1.39 | 43.06 | 5.53 | 50.36 | 1.05 |
| | Coherence | 49.58 | 26.27 | 22.82 | 1.32 | 57.23 | 20.34 | 21.04 | 1.39 | 65.16 | 11.99 | 21.36 | 1.49 |
| | Plot Conclusion | 40.04 | 22.51 | 36.43 | 1.02 | 51.68 | 20.06 | 26.84 | 1.41 | 55.8 | 14.89 | 28.28 | 1.03 |
| **Predictive Maintenance** | Depth of Analysis | 51.2 | 27.15 | 20.24 | 1.41 | 60.01 | 19.68 | 19.03 | 1.24 | 64.82 | 12.05 | 22.04 | 1.1 |
| | Relevance To Goal | 33.42 | 15.66 | 48.85 | 2.07 | 45.36 | 10.44 | 43.04 | 1.16 | 53.34 | 5.12 | 40.43 | 1.11 |
| | Persona Consistency | 26.9 | 9.62 | 61.0 | 2.48 | 37.31 | 7.65 | 53.71 | 1.33 | 40.77 | 4.23 | 53.75 | 1.25 |
| | Coherence | 50.41 | 27.59 | 19.11 | 2.89 | 59.11 | 19.56 | 20.22 | 1.11 | 64.15 | 13.2 | 21.48 | 1.17 |
| | Plot Conclusion | 42.25 | 24.31 | 30.62 | 2.81 | 51.93 | 19.57 | 27.26 | 1.25 | 54.88 | 15.15 | 28.74 | 1.23 |
| **Cohort Analysis** | Depth of Analysis | 47.81 | 28.35 | 21.65 | 2.19 | 56.98 | 20.82 | 21.17 | 1.03 | 63.61 | 12.73 | 22.3 | 1.36 |
| | Relevance To Goal | 31.86 | 19.27 | 47.7 | 1.17 | 44.6 | 7.54 | 46.58 | 1.28 | 50.54 | 4.85 | 43.16 | 1.44 |
| | Persona Consistency | 26.45 | 11.0 | 60.77 | 1.79 | 36.88 | 7.81 | 54.0 | 1.31 | 42.81 | 4.3 | 51.64 | 1.25 |
| | Coherence | 47.94 | 29.18 | 20.22 | 2.66 | 58.5 | 21.26 | 18.77 | 1.48 | 63.25 | 13.48 | 22.13 | 1.14 |
| | Plot Conclusion | 38.71 | 23.32 | 36.54 | 1.43 | 50.69 | 19.64 | 28.45 | 1.22 | 55.08 | 15.03 | 28.77 | 1.13 |
| **Anomaly Detection** | Depth of Analysis | 50.72 | 28.28 | 19.9 | 1.1 | 58.43 | 19.15 | 21.14 | 1.29 | 64.3 | 13.34 | 21.0 | 1.35 |
| | Relevance To Goal | 33.98 | 15.93 | 47.94 | 2.15 | 46.86 | 10.18 | 41.6 | 1.36 | 51.7 | 6.78 | 40.48 | 1.04 |
| | Persona Consistency | 25.44 | 9.94 | 63.54 | 1.08 | 37.29 | 7.9 | 53.78 | 1.03 | 42.36 | 4.08 | 52.32 | 1.24 |
| | Coherence | 48.5 | 30.21 | 19.15 | 2.14 | 60.68 | 19.43 | 18.55 | 1.34 | 64.8 | 10.22 | 23.91 | 1.07 |

Table 22: Insight-wise SCORER comparison between AGENTADA W Skill and Other agents (Part 2).

| Task | Rubric | InfiAgent | | | | MetaGPT | | | | GPT-4o | | | |
|---|---|---|---|---|---|---|---|---|---|---|---|---|---|
| | | WA | WO | T | N | WA | WO | T | N | WA | WO | T | N |
| Sentiment Analysis | Depth of Analysis | 56.35 | 21.88 | 20.74 | 1.03 | 57.52 | 22.34 | 18.66 | 1.48 | 60.75 | 17.74 | 20.2 | 1.31 |
| | Relevance To Goal | 39.04 | 11.13 | 48.35 | 1.48 | 41.42 | 10.39 | 47.1 | 1.1 | 51.07 | 7.61 | 40.13 | 1.19 |
| | Persona Consistency | 32.79 | 9.18 | 56.8 | 1.23 | 36.3 | 7.25 | 55.42 | 1.03 | 41.02 | 4.28 | 53.21 | 1.49 |
| | Coherence | 57.53 | 21.1 | 19.97 | 1.41 | 57.15 | 20.05 | 21.41 | 1.39 | 59.65 | 18.26 | 20.76 | 1.32 |
| | Plot Conclusion | 46.3 | 23.51 | 28.8 | 1.39 | 50.62 | 19.33 | 28.64 | 1.41 | 53.32 | 14.51 | 30.87 | 1.3 |
| A/B Testing | Depth of Analysis | 55.43 | 22.72 | 20.62 | 1.23 | 56.98 | 20.37 | 21.19 | 1.46 | 60.85 | 15.61 | 22.37 | 1.17 |
| | Relevance To Goal | 39.8 | 13.58 | 45.55 | 1.08 | 42.48 | 11.81 | 44.59 | 1.12 | 47.99 | 10.04 | 40.56 | 1.41 |
| | Persona Consistency | 31.5 | 8.7 | 58.57 | 1.23 | 35.24 | 7.35 | 56.25 | 1.16 | 39.71 | 4.62 | 54.51 | 1.15 |
| | Coherence | 55.7 | 23.19 | 19.7 | 1.41 | 58.19 | 18.83 | 21.93 | 1.05 | 61.85 | 16.15 | 20.82 | 1.18 |
| | Plot Conclusion | 50.58 | 22.07 | 26.14 | 1.21 | 51.43 | 18.61 | 28.77 | 1.19 | 51.05 | 16.99 | 30.69 | 1.27 |
| Forecasting | Depth of Analysis | 54.38 | 25.15 | 19.28 | 1.19 | 59.29 | 21.21 | 18.25 | 1.25 | 59.25 | 17.77 | 21.63 | 1.35 |
| | Relevance To Goal | 41.46 | 11.22 | 46.28 | 1.04 | 42.8 | 10.84 | 44.91 | 1.45 | 47.08 | 8.46 | 43.03 | 1.43 |
| | Persona Consistency | 30.81 | 11.44 | 56.35 | 1.39 | 38.01 | 8.82 | 51.92 | 1.24 | 40.49 | 6.4 | 51.72 | 1.39 |
| | Coherence | 57.65 | 21.34 | 19.87 | 1.14 | 55.65 | 21.39 | 21.7 | 1.25 | 62.0 | 15.37 | 21.32 | 1.31 |
| | Plot Conclusion | 47.89 | 22.16 | 28.65 | 1.3 | 49.57 | 20.41 | 28.79 | 1.23 | 53.92 | 16.96 | 28.07 | 1.05 |
| Basic Data Analysis | Depth of Analysis | 59.2 | 21.44 | 18.25 | 1.11 | 58.37 | 22.34 | 18.16 | 1.14 | 62.34 | 15.91 | 20.45 | 1.3 |
| | Relevance To Goal | 38.66 | 13.13 | 46.76 | 1.45 | 43.93 | 11.73 | 43.14 | 1.2 | 47.14 | 8.47 | 43.39 | 1.0 |
| | Persona Consistency | 32.81 | 9.86 | 55.89 | 1.44 | 36.21 | 7.41 | 55.12 | 1.26 | 40.61 | 6.9 | 51.11 | 1.38 |
| | Coherence | 57.66 | 20.78 | 20.4 | 1.16 | 56.76 | 21.86 | 20.1 | 1.28 | 60.89 | 16.77 | 20.94 | 1.4 |
| | Plot Conclusion | 50.37 | 20.62 | 27.92 | 1.1 | 51.3 | 18.31 | 29.36 | 1.03 | 52.91 | 17.29 | 28.63 | 1.18 |
| Recommendation Systems | Depth of Analysis | 57.64 | 22.57 | 18.34 | 1.45 | 57.91 | 21.64 | 19.33 | 1.12 | 61.55 | 14.59 | 22.49 | 1.36 |
| | Relevance To Goal | 38.79 | 10.45 | 49.75 | 1.01 | 41.19 | 10.0 | 47.63 | 1.18 | 49.92 | 7.19 | 41.72 | 1.16 |
| | Persona Consistency | 31.18 | 11.75 | 55.89 | 1.18 | 36.38 | 8.52 | 54.09 | 1.01 | 40.21 | 6.06 | 52.51 | 1.22 |
| | Coherence | 56.15 | 21.87 | 20.71 | 1.27 | 58.57 | 21.72 | 18.53 | 1.18 | 64.08 | 15.13 | 19.53 | 1.25 |
| | Plot Conclusion | 51.06 | 21.89 | 26.04 | 1.01 | 49.01 | 21.4 | 28.31 | 1.28 | 53.44 | 17.11 | 28.14 | 1.32 |
| Dashboard Summary | Depth of Analysis | 53.74 | 24.96 | 20.2 | 1.1 | 57.62 | 23.27 | 17.93 | 1.18 | 60.73 | 16.04 | 22.22 | 1.02 |
| | Relevance To Goal | 39.82 | 14.45 | 44.33 | 1.4 | 42.47 | 9.45 | 46.85 | 1.23 | 48.56 | 7.75 | 42.54 | 1.16 |
| | Persona Consistency | 33.23 | 11.37 | 54.25 | 1.15 | 35.69 | 6.95 | 56.15 | 1.2 | 40.89 | 6.37 | 51.55 | 1.19 |
| | Coherence | 53.36 | 24.63 | 20.56 | 1.45 | 58.42 | 20.6 | 19.68 | 1.31 | 64.14 | 15.21 | 19.64 | 1.01 |
| | Plot Conclusion | 48.6 | 21.81 | 28.58 | 1.01 | 50.04 | 22.21 | 26.55 | 1.2 | 53.53 | 15.54 | 29.58 | 1.35 |
| Customer Segmentation | Depth of Analysis | 55.25 | 23.36 | 20.16 | 1.23 | 55.53 | 21.02 | 22.07 | 1.38 | 63.07 | 15.17 | 20.33 | 1.43 |
| | Relevance To Goal | 37.32 | 13.34 | 48.0 | 1.34 | 43.47 | 9.55 | 45.63 | 1.24 | 49.01 | 7.26 | 42.44 | 1.28 |
| | Persona Consistency | 34.38 | 7.61 | 56.58 | 1.43 | 35.01 | 7.28 | 56.69 | 1.02 | 42.16 | 7.16 | 49.57 | 1.1 |
| | Coherence | 55.04 | 23.38 | 20.3 | 1.28 | 59.1 | 20.38 | 19.12 | 1.39 | 62.22 | 18.01 | 18.74 | 1.02 |
| | Plot Conclusion | 50.07 | 23.04 | 25.59 | 1.3 | 49.43 | 19.13 | 30.36 | 1.08 | 52.51 | 16.86 | 29.42 | 1.22 |
| Network Analysis | Depth of Analysis | 58.12 | 22.75 | 17.71 | 1.42 | 57.14 | 20.87 | 20.85 | 1.14 | 60.76 | 17.25 | 20.96 | 1.03 |
| | Relevance To Goal | 42.33 | 10.89 | 45.51 | 1.26 | 42.92 | 8.45 | 47.18 | 1.45 | 47.5 | 9.51 | 41.52 | 1.48 |
| | Persona Consistency | 31.34 | 7.47 | 59.87 | 1.31 | 37.61 | 6.97 | 54.31 | 1.11 | 40.3 | 7.11 | 51.1 | 1.49 |
| | Coherence | 56.03 | 21.67 | 20.86 | 1.44 | 58.54 | 21.64 | 18.55 | 1.27 | 59.9 | 17.49 | 21.29 | 1.32 |
| | Plot Conclusion | 49.2 | 22.14 | 27.48 | 1.18 | 50.69 | 18.21 | 29.68 | 1.42 | 52.37 | 15.67 | 30.56 | 1.41 |
| Association Rule Mining | Depth of Analysis | 56.82 | 22.32 | 19.39 | 1.47 | 58.91 | 19.25 | 20.4 | 1.44 | 61.37 | 16.12 | 21.47 | 1.04 |
| | Relevance To Goal | 38.08 | 12.89 | 47.65 | 1.39 | 42.89 | 10.65 | 45.27 | 1.19 | 46.56 | 8.0 | 44.42 | 1.02 |
| | Persona Consistency | 33.5 | 11.06 | 54.28 | 1.16 | 37.47 | 7.44 | 53.86 | 1.24 | 39.98 | 7.21 | 51.71 | 1.1 |
| | Coherence | 54.26 | 24.48 | 20.2 | 1.06 | 58.31 | 19.7 | 20.88 | 1.12 | 62.95 | 15.9 | 19.99 | 1.15 |
| | Plot Conclusion | 47.58 | 21.85 | 29.27 | 1.3 | 50.66 | 19.78 | 28.15 | 1.41 | 53.75 | 16.25 | 28.53 | 1.47 |
| Predictive Maintenance | Depth of Analysis | 56.52 | 23.42 | 18.87 | 1.19 | 57.65 | 20.39 | 20.77 | 1.19 | 60.38 | 17.66 | 20.76 | 1.2 |
| | Relevance To Goal | 37.33 | 14.98 | 46.57 | 1.12 | 41.16 | 10.92 | 46.46 | 1.46 | 48.5 | 6.87 | 43.33 | 1.29 |
| | Persona Consistency | 33.3 | 11.02 | 54.52 | 1.16 | 37.09 | 7.43 | 54.14 | 1.34 | 39.83 | 7.03 | 52.11 | 1.03 |
| | Coherence | 54.86 | 23.22 | 20.52 | 1.4 | 58.46 | 18.64 | 21.7 | 1.2 | 60.56 | 18.14 | 19.9 | 1.41 |
| | Plot Conclusion | 47.74 | 23.84 | 27.18 | 1.24 | 51.74 | 18.82 | 28.04 | 1.4 | 54.68 | 15.22 | 28.64 | 1.47 |
| Cohort Analysis | Depth of Analysis | 56.4 | 21.0 | 21.55 | 1.05 | 58.06 | 22.08 | 18.37 | 1.49 | 62.03 | 17.33 | 19.17 | 1.47 |
| | Relevance To Goal | 38.71 | 15.47 | 44.64 | 1.18 | 42.77 | 9.12 | 46.73 | 1.38 | 47.44 | 7.41 | 43.77 | 1.38 |
| | Persona Consistency | 33.23 | 11.19 | 54.52 | 1.05 | 36.86 | 7.26 | 54.46 | 1.42 | 39.6 | 7.51 | 51.76 | 1.14 |
| | Coherence | 56.38 | 21.86 | 20.45 | 1.3 | 57.73 | 19.66 | 21.25 | 1.36 | 62.7 | 17.56 | 18.54 | 1.21 |
| | Plot Conclusion | 48.55 | 23.32 | 26.73 | 1.4 | 50.74 | 20.21 | 27.82 | 1.23 | 52.6 | 16.23 | 30.16 | 1.02 |
| Attribution Modeling | Depth of Analysis | 55.85 | 21.96 | 20.71 | 1.48 | 56.15 | 22.69 | 19.97 | 1.19 | 62.31 | 17.24 | 19.29 | 1.16 |
| | Relevance To Goal | 38.04 | 13.92 | 46.79 | 1.25 | 42.07 | 10.9 | 45.62 | 1.41 | 47.96 | 7.96 | 43.03 | 1.05 |
| | Persona Consistency | 32.76 | 10.56 | 55.25 | 1.43 | 37.66 | 8.49 | 52.52 | 1.33 | 39.69 | 7.64 | 51.58 | 1.09 |
| | Coherence | 54.26 | 24.48 | 20.2 | 1.06 | 58.31 | 19.7 | 20.88 | 1.12 | 62.95 | 15.9 | 19.99 | 1.15 |
| | Plot Conclusion | 47.58 | 21.85 | 29.27 | 1.3 | 50.66 | 19.78 | 28.15 | 1.41 | 53.75 | 16.25 | 28.53 | 1.47 |
| Anomaly Detection | Depth of Analysis | 57.38 | 21.21 | 20.14 | 1.27 | 56.81 | 21.12 | 20.89 | 1.19 | 60.25 | 17.28 | 21.43 | 1.05 |
| | Relevance To Goal | 41.54 | 10.76 | 46.44 | 1.26 | 42.57 | 10.37 | 45.78 | 1.28 | 49.28 | 7.47 | 42.16 | 1.09 |
| | Persona Consistency | 35.52 | 7.81 | 55.18 | 1.49 | 36.65 | 6.63 | 55.25 | 1.47 | 41.57 | 6.57 | 50.62 | 1.24 |
| | Coherence | 56.82 | 21.98 | 19.77 | 1.42 | 56.47 | 21.0 | 21.37 | 1.16 | 62.75 | 16.59 | 19.57 | 1.09 |

Table 23: Insight-wise SCORER comparison between AGENTADA W Skill and Other agents (Part 3).

| Task | Rubric | w/o skill | | | | Poirot | | | | Pandas | | | |
|---|---|---|---|---|---|---|---|---|---|---|---|---|---|
| | | WA | WO | T | N | WA | WO | T | N | WA | WO | T | N |
| **Feature Importance Ranking** | *Depth of Analysis* | **47.15** | 27.77 | 22.7 | 2.38 | **61.24** | 17.98 | 19.32 | 1.45 | **64.74** | 13.48 | 20.49 | 1.3 |
| | *Relevance To Goal* | 33.74 | 14.27 | **49.79** | 2.2 | **44.76** | 11.29 | 42.7 | 1.26 | **50.02** | 6.75 | 42.09 | 1.14 |
| | *Persona Consistency* | 24.21 | 11.89 | **62.73** | 1.17 | 36.47 | 5.59 | **56.59** | 1.34 | 43.27 | 4.3 | **51.26** | 1.17 |
| | *Coherence* | **51.84** | 26.4 | 19.43 | 2.33 | **57.83** | 20.08 | 20.95 | 1.14 | **63.25** | 12.97 | 22.53 | 1.25 |
| | *Plot Conclusion* | 41.26 | 25.92 | 31.12 | 1.71 | **52.76** | 19.5 | 26.24 | 1.5 | **58.14** | 13.86 | 26.92 | 1.08 |
| **Geospatial Analysis** | *Depth of Analysis* | 49.46 | 29.33 | 19.88 | 1.34 | **58.55** | 20.9 | 19.39 | 1.16 | **61.84** | 12.88 | 23.79 | 1.49 |
| | *Relevance To Goal* | 32.65 | 15.85 | **49.72** | 1.78 | 43.72 | 10.67 | **44.38** | 1.23 | **53.03** | 4.85 | 40.78 | 1.34 |
| | *Persona Consistency* | 25.41 | 9.15 | **62.92** | 2.52 | 38.44 | 8.45 | **51.76** | 1.35 | 40.86 | 4.77 | **53.31** | 1.05 |
| | *Coherence* | 50.03 | 26.78 | 21.78 | 1.41 | **59.0** | 19.18 | 20.5 | 1.33 | **65.92** | 10.33 | 22.54 | 1.21 |
| | *Plot Conclusion* | 43.26 | 23.07 | 31.36 | 2.31 | 50.55 | 19.89 | 28.24 | 1.32 | **56.82** | 14.21 | 27.63 | 1.34 |
| **Causality** | *Depth of Analysis* | 49.01 | 28.56 | 19.81 | 2.62 | **60.9** | 17.07 | 20.96 | 1.07 | **64.03** | 11.97 | 22.88 | 1.12 |
| | *Relevance To Goal* | 31.87 | 16.22 | **50.16** | 1.75 | **44.58** | 10.81 | 43.61 | 1.0 | **49.99** | 7.98 | 40.76 | 1.27 |
| | *Persona Consistency* | 24.91 | 9.56 | **62.64** | 2.89 | 38.73 | 6.6 | **53.19** | 1.48 | 40.53 | 5.51 | **52.96** | 1.0 |
| | *Coherence* | **46.76** | 28.62 | 22.58 | 2.04 | **57.78** | 19.14 | 21.82 | 1.26 | **62.1** | 13.45 | 22.98 | 1.47 |
| | *Plot Conclusion* | 42.12 | 22.29 | 33.78 | 1.81 | 50.42 | 20.12 | 28.16 | 1.3 | 54.24 | 14.76 | 29.59 | 1.4 |
| **Causality Analysis** | *Depth of Analysis* | 47.79 | 28.79 | 20.45 | 2.97 | 57.55 | 21.73 | 19.29 | 1.43 | **64.08** | 12.12 | 22.65 | 1.15 |
| | *Relevance To Goal* | 31.39 | 18.34 | **48.9** | 1.37 | 47.44 | 8.13 | 42.98 | 1.45 | **50.27** | 7.08 | 41.37 | 1.28 |
| | *Persona Consistency* | 24.26 | 11.39 | **63.13** | 1.22 | 37.95 | 6.25 | **54.52** | 1.27 | 42.1 | 3.91 | **52.97** | 1.02 |
| | *Coherence* | 50.93 | 27.15 | 20.49 | 1.44 | **59.6** | 19.93 | 19.33 | 1.14 | **65.07** | 11.37 | 22.25 | 1.31 |
| | *Plot Conclusion* | 40.53 | 22.51 | 34.64 | 2.33 | 52.29 | 18.89 | 27.61 | 1.21 | 57.36 | 13.57 | 27.69 | 1.37 |
| **Logs Clustering** | *Depth of Analysis* | 47.83 | 28.31 | 22.34 | 1.53 | **61.25** | 17.43 | 20.01 | 1.31 | **62.84** | 12.6 | 23.13 | 1.42 |
| | *Relevance To Goal* | 35.12 | 14.91 | **47.68** | 2.29 | 44.69 | 9.47 | 44.44 | 1.4 | **49.27** | 5.93 | 43.77 | 1.03 |
| | *Persona Consistency* | 24.74 | 9.01 | **64.2** | 2.05 | 37.31 | 7.15 | **54.46** | 1.08 | 43.7 | 5.62 | **49.56** | 1.12 |
| | *Coherence* | 50.78 | 28.82 | 18.79 | 1.61 | **60.08** | 17.76 | 20.83 | 1.33 | **63.26** | 12.92 | 22.37 | 1.45 |
| | *Plot Conclusion* | 38.73 | 23.96 | 34.59 | 2.72 | 53.26 | 19.28 | 26.43 | 1.04 | **58.75** | 12.37 | 27.72 | 1.16 |
| **Time Series Decomposition** | *Depth of Analysis* | 47.64 | 26.81 | 23.16 | 2.39 | **60.36** | 20.36 | 18.25 | 1.03 | **62.23** | 13.0 | 23.32 | 1.45 |
| | *Relevance To Goal* | 34.73 | 13.35 | **50.46** | 1.46 | 44.21 | 9.18 | **45.52** | 1.09 | **50.47** | 6.39 | 41.93 | 1.21 |
| | *Persona Consistency* | 28.59 | 8.05 | **60.94** | 2.41 | 37.7 | 6.94 | **54.31** | 1.06 | 43.78 | 4.67 | **50.34** | 1.21 |
| | *Coherence* | 47.95 | 27.85 | 22.34 | 1.86 | 57.62 | 20.11 | 21.13 | 1.14 | **63.71** | 12.55 | 22.68 | 1.06 |
| | *Plot Conclusion* | 42.56 | 23.08 | 32.34 | 2.02 | **51.9** | 21.16 | 25.73 | 1.21 | **57.78** | 12.04 | 29.04 | 1.14 |
| **Principal Component Analysis** | *Depth of Analysis* | 51.75 | 26.85 | 20.26 | 1.14 | 58.91 | 21.14 | 18.84 | 1.12 | **66.94** | 10.77 | 21.19 | 1.1 |
| | *Relevance To Goal* | 32.15 | 16.95 | **48.15** | 2.75 | 47.09 | 9.07 | 42.7 | 1.13 | **50.83** | 8.29 | 39.49 | 1.39 |
| | *Persona Consistency* | 24.83 | 8.88 | **65.19** | 1.09 | 38.97 | 6.93 | **52.8** | 1.31 | 44.4 | 3.46 | **50.96** | 1.18 |
| | *Coherence* | 50.67 | 28.43 | 19.16 | 1.74 | **58.1** | 20.69 | 19.95 | 1.26 | **63.3** | 11.08 | 24.3 | 1.33 |
| | *Plot Conclusion* | 39.45 | 22.89 | 35.41 | 2.25 | 50.48 | 20.93 | 27.38 | 1.21 | **55.16** | 15.14 | 28.59 | 1.11 |
| **Correlation Analysis** | *Depth of Analysis* | 46.06 | 28.56 | 23.31 | 2.07 | **61.19** | 18.37 | 19.08 | 1.35 | **63.54** | 12.89 | 22.56 | 1.01 |
| | *Relevance To Goal* | 34.03 | 14.79 | **49.78** | 1.4 | 44.64 | 7.68 | **46.58** | 1.1 | **49.49** | 7.58 | 41.78 | 1.16 |
| | *Persona Consistency* | 27.52 | 12.57 | **57.32** | 2.59 | 36.88 | 9.57 | **52.36** | 1.19 | 43.08 | 6.06 | **49.47** | 1.39 |
| | *Coherence* | 50.35 | 26.3 | 20.57 | 2.79 | 57.57 | 21.36 | 19.64 | 1.43 | **63.55** | 11.84 | 23.35 | 1.27 |
| | *Plot Conclusion* | 43.48 | 23.34 | 31.76 | 1.42 | 51.77 | 19.47 | 27.7 | 1.06 | 57.24 | 13.78 | 27.96 | 1.02 |
| **Knowledge Base** | *Depth of Analysis* | 45.87 | 29.11 | 22.07 | 2.95 | 59.39 | 18.77 | 20.36 | 1.48 | **63.45** | 12.8 | 22.49 | 1.26 |
| | *Relevance To Goal* | 34.63 | 16.63 | **46.12** | 2.62 | 44.94 | 8.57 | **45.06** | 1.43 | **48.43** | 8.08 | 42.26 | 1.23 |
| | *Persona Consistency* | 25.34 | 12.64 | **60.33** | 1.7 | 37.82 | 7.52 | **53.31** | 1.35 | 41.85 | 5.89 | **50.97** | 1.29 |
| | *Coherence* | 48.66 | 27.35 | 21.4 | 2.59 | **60.86** | 18.65 | 19.22 | 1.27 | **64.26** | 11.01 | 23.48 | 1.25 |
| | *Plot Conclusion* | 39.96 | 22.42 | 35.34 | 2.48 | 53.75 | 18.31 | 26.83 | 1.1 | **56.25** | 13.89 | 28.41 | 1.45 |
| **Huge Table Analysis** | *Depth of Analysis* | 47.24 | 28.24 | 22.9 | 1.62 | 58.15 | 19.92 | 20.76 | 1.17 | **65.29** | 11.18 | 22.29 | 1.24 |
| | *Relevance To Goal* | 32.32 | 20.64 | **45.5** | 1.54 | 45.54 | 8.63 | 44.7 | 1.14 | **48.04** | 8.09 | 42.39 | 1.48 |
| | *Persona Consistency* | 26.69 | 11.65 | **60.05** | 1.61 | 39.18 | 8.62 | **51.12** | 1.09 | 40.34 | 5.14 | **53.48** | 1.04 |
| | *Coherence* | 48.5 | 30.35 | 19.28 | 1.87 | 58.4 | 19.35 | 21.1 | 1.15 | **64.85** | 11.27 | 22.44 | 1.44 |
| | *Plot Conclusion* | 43.08 | 24.03 | 30.6 | 2.29 | 50.54 | 20.98 | 27.22 | 1.25 | 55.68 | 13.44 | 29.75 | 1.13 |
| **Topic Modeling** | *Depth of Analysis* | 46.45 | 28.61 | 23.01 | 1.93 | **61.99** | 18.33 | 18.43 | 1.25 | **63.95** | 12.14 | 22.53 | 1.37 |
| | *Relevance To Goal* | 34.39 | 13.11 | **50.55** | 1.95 | 42.82 | 9.77 | **46.37** | 1.04 | **52.44** | 5.07 | 41.45 | 1.04 |
| | *Persona Consistency* | 24.6 | 12.94 | **59.61** | 2.86 | 40.21 | 5.5 | **53.24** | 1.05 | 41.72 | 5.18 | **51.69** | 1.4 |
| | *Coherence* | 51.07 | 26.3 | 20.95 | 1.68 | 56.67 | 20.39 | 21.64 | 1.3 | **64.99** | 10.91 | 22.9 | 1.2 |
| | *Plot Conclusion* | 44.65 | 23.25 | 30.95 | 1.15 | 51.92 | 19.05 | 28.03 | 1.0 | **57.37** | 12.69 | 28.78 | 1.16 |
| **Market Analysis** | *Depth of Analysis* | 50.72 | 25.39 | 21.42 | 2.48 | 59.68 | 19.19 | 19.73 | 1.4 | **62.82** | 13.91 | 22.08 | 1.19 |
| | *Relevance To Goal* | 34.16 | 13.78 | **50.17** | 1.89 | 43.45 | 8.07 | **47.18** | 1.3 | **50.62** | 7.93 | 40.22 | 1.23 |
| | *Persona Consistency* | 27.15 | 11.31 | **58.6** | 2.94 | 39.23 | 8.65 | **50.87** | 1.25 | 42.31 | 3.82 | **52.53** | 1.33 |
| | *Coherence* | 48.96 | 29.39 | 19.28 | 2.37 | **59.09** | 18.26 | 21.36 | 1.29 | **65.87** | 11.58 | 21.27 | 1.28 |
| | *Plot Conclusion* | 39.61 | 23.26 | 34.61 | 2.52 | 50.65 | 20.86 | 27.3 | 1.19 | 56.14 | 13.1 | 29.54 | 1.22 |
| **Data Imputation** | *Depth of Analysis* | 49.61 | 25.98 | 21.83 | 2.58 | **61.13** | 20.32 | 17.54 | 1.01 | **63.91** | 12.65 | 22.44 | 1.0 |
| | *Relevance To Goal* | 30.56 | 16.61 | **50.2** | 2.63 | 46.09 | 9.12 | 43.59 | 1.2 | **50.86** | 7.66 | 40.36 | 1.13 |
| | *Persona Consistency* | 24.94 | 10.08 | **62.38** | 2.6 | 37.12 | 6.58 | **55.17** | 1.13 | 43.52 | 5.68 | **49.78** | 1.02 |
| | *Coherence* | **47.49** | 28.71 | 22.45 | 1.35 | **60.45** | 19.51 | 18.9 | 1.14 | **64.91** | 12.18 | 21.54 | 1.37 |
| | *Plot Conclusion* | **39.1** | 24.67 | 34.3 | 1.93 | 50.15 | 19.7 | 28.94 | 1.21 | **56.22** | 12.62 | 30.09 | 1.07 |
| **Multi-table Search** | *Depth of Analysis* | 50.21 | 27.07 | 19.79 | 2.93 | 58.09 | 21.92 | 18.97 | 1.02 | **63.7** | 10.66 | 24.4 | 1.24 |
| | *Relevance To Goal* | 30.92 | 21.18 | **46.89** | 1.01 | 47.56 | 7.61 | 43.5 | 1.33 | **52.71** | 7.67 | 38.15 | 1.47 |
| | *Persona Consistency* | 27.18 | 7.81 | **62.39** | 2.61 | 38.67 | 7.32 | **52.51** | 1.5 | 42.24 | 3.48 | **53.16** | 1.12 |
| | *Coherence* | **48.7** | 26.96 | 21.43 | 2.91 | 56.97 | 20.03 | 21.7 | 1.3 | **62.91** | 12.88 | 22.91 | 1.3 |
| | *Plot Conclusion* | 43.31 | 24.43 | 30.27 | 1.99 | 50.04 | 19.85 | 28.95 | 1.16 | **55.6** | 13.68 | 29.25 | 1.48 |

Table 24: Insight-wise SCORER comparison between AGENTADA W Skill and Other agents (Part 4).

| Task | Rubric | InfiAgent | | | | MetaGPT | | | | GPT-4o | | | |
|---|---|---|---|---|---|---|---|---|---|---|---|---|---|
| | | WA | WO | T | N | WA | WO | T | N | WA | WO | T | N |
| **Feature Importance Ranking** | Depth of Analysis | 57.37 | 20.71 | 20.47 | 1.45 | 59.39 | 21.46 | 17.93 | 1.22 | 61.58 | 16.26 | 20.92 | 1.25 |
| | Relevance To Goal | 37.16 | 13.83 | 47.78 | 1.23 | 42.23 | 12.44 | 43.87 | 1.46 | 49.15 | 9.16 | 40.21 | 1.48 |
| | Persona Consistency | 32.99 | 8.93 | 56.68 | 1.4 | 34.61 | 10.0 | 54.2 | 1.19 | 42.83 | 5.29 | 50.53 | 1.35 |
| | Coherence | 55.62 | 20.92 | 22.13 | 1.33 | 57.76 | 21.15 | 19.95 | 1.14 | 60.28 | 16.92 | 21.63 | 1.17 |
| | Plot Conclusion | 49.45 | 23.65 | 25.87 | 1.03 | 50.89 | 21.51 | 26.52 | 1.09 | 55.61 | 14.71 | 28.54 | 1.14 |
| **Geospatial Analysis** | Depth of Analysis | 53.95 | 24.38 | 20.41 | 1.26 | 59.88 | 20.7 | 18.17 | 1.25 | 61.66 | 15.85 | 21.19 | 1.3 |
| | Relevance To Goal | 39.05 | 13.35 | 46.16 | 1.43 | 43.94 | 10.35 | 44.65 | 1.06 | 45.1 | 10.35 | 43.45 | 1.1 |
| | Persona Consistency | 31.31 | 7.77 | 59.67 | 1.25 | 37.3 | 7.13 | 54.48 | 1.08 | 39.67 | 7.61 | 51.69 | 1.02 |
| | Coherence | 58.1 | 22.17 | 18.46 | 1.27 | 58.33 | 20.19 | 20.24 | 1.25 | 62.23 | 16.1 | 20.31 | 1.36 |
| | Plot Conclusion | 47.92 | 22.79 | 28.02 | 1.27 | 50.97 | 20.85 | 26.83 | 1.35 | 54.54 | 14.69 | 29.49 | 1.28 |
| **Causality** | Depth of Analysis | 55.96 | 23.55 | 19.14 | 1.36 | 58.78 | 20.56 | 19.23 | 1.43 | 62.37 | 16.83 | 19.49 | 1.31 |
| | Relevance To Goal | 41.11 | 11.31 | 46.37 | 1.21 | 40.17 | 12.04 | 46.67 | 1.11 | 45.96 | 9.89 | 43.07 | 1.08 |
| | Persona Consistency | 32.65 | 9.06 | 56.97 | 1.32 | 35.34 | 7.1 | 56.32 | 1.24 | 39.02 | 6.57 | 53.34 | 1.07 |
| | Coherence | 55.44 | 22.2 | 21.02 | 1.35 | 56.76 | 19.46 | 22.47 | 1.31 | 61.48 | 17.09 | 20.07 | 1.36 |
| | Plot Conclusion | 51.17 | 22.53 | 25.17 | 1.13 | 50.85 | 20.05 | 27.61 | 1.49 | 54.0 | 15.8 | 28.81 | 1.4 |
| **Causality Analysis** | Depth of Analysis | 55.4 | 24.96 | 18.56 | 1.08 | 60.26 | 20.63 | 18.01 | 1.09 | 62.57 | 16.66 | 19.69 | 1.08 |
| | Relevance To Goal | 40.39 | 12.01 | 46.44 | 1.16 | 43.19 | 12.21 | 43.27 | 1.33 | 50.75 | 6.61 | 41.25 | 1.39 |
| | Persona Consistency | 34.16 | 7.28 | 57.17 | 1.39 | 35.59 | 8.2 | 55.2 | 1.02 | 39.31 | 5.06 | 54.13 | 1.5 |
| | Coherence | 56.39 | 24.24 | 18.28 | 1.09 | 57.54 | 20.32 | 20.93 | 1.21 | 60.23 | 18.55 | 19.81 | 1.41 |
| | Plot Conclusion | 49.96 | 21.93 | 26.85 | 1.27 | 51.94 | 19.23 | 27.54 | 1.3 | 52.53 | 15.94 | 30.36 | 1.16 |
| **Logs Clustering** | Depth of Analysis | 58.02 | 22.86 | 17.75 | 1.37 | 59.44 | 18.85 | 20.53 | 1.18 | 61.54 | 16.24 | 21.19 | 1.02 |
| | Relevance To Goal | 40.63 | 11.45 | 46.86 | 1.06 | 39.68 | 9.38 | 49.76 | 1.18 | 48.43 | 9.98 | 40.58 | 1.01 |
| | Persona Consistency | 32.38 | 11.56 | 55.03 | 1.04 | 35.87 | 8.53 | 54.51 | 1.1 | 39.15 | 6.99 | 52.69 | 1.17 |
| | Coherence | 56.74 | 21.06 | 21.01 | 1.19 | 56.55 | 20.22 | 22.07 | 1.16 | 63.02 | 15.67 | 20.08 | 1.24 |
| | Plot Conclusion | 48.3 | 23.34 | 27.21 | 1.15 | 50.9 | 19.03 | 28.95 | 1.12 | 51.56 | 16.94 | 30.05 | 1.45 |
| **Time Series Decomposition** | Depth of Analysis | 56.11 | 24.47 | 17.98 | 1.44 | 59.29 | 21.56 | 17.94 | 1.21 | 63.53 | 15.06 | 20.14 | 1.27 |
| | Relevance To Goal | 39.09 | 14.88 | 45.0 | 1.03 | 45.19 | 9.34 | 44.46 | 1.01 | 49.06 | 9.33 | 40.16 | 1.45 |
| | Persona Consistency | 33.82 | 10.0 | 55.16 | 1.02 | 38.27 | 7.04 | 53.31 | 1.38 | 39.76 | 7.07 | 51.79 | 1.39 |
| | Coherence | 52.92 | 24.43 | 21.16 | 1.49 | 57.25 | 21.15 | 20.26 | 1.34 | 63.62 | 15.64 | 19.55 | 1.19 |
| | Plot Conclusion | 49.86 | 21.04 | 27.91 | 1.19 | 50.17 | 21.45 | 27.04 | 1.35 | 54.05 | 16.83 | 27.93 | 1.19 |
| **Principal Component Analysis** | Depth of Analysis | 55.17 | 23.77 | 19.66 | 1.4 | 56.86 | 20.41 | 21.53 | 1.2 | 60.23 | 18.08 | 20.26 | 1.43 |
| | Relevance To Goal | 37.37 | 10.77 | 50.4 | 1.46 | 43.07 | 10.18 | 45.71 | 1.04 | 47.46 | 10.66 | 40.86 | 1.02 |
| | Persona Consistency | 32.83 | 8.73 | 57.33 | 1.11 | 34.87 | 9.16 | 54.96 | 1.0 | 40.66 | 4.59 | 53.64 | 1.12 |
| | Coherence | 55.5 | 22.46 | 20.84 | 1.2 | 57.99 | 20.98 | 19.59 | 1.44 | 61.14 | 16.75 | 20.81 | 1.29 |
| | Plot Conclusion | 48.4 | 22.59 | 27.97 | 1.04 | 49.85 | 20.39 | 28.43 | 1.33 | 54.17 | 15.12 | 29.33 | 1.38 |
| **Correlation Analysis** | Depth of Analysis | 55.59 | 21.11 | 22.0 | 1.29 | 57.96 | 19.84 | 20.73 | 1.47 | 61.36 | 16.22 | 20.99 | 1.44 |
| | Relevance To Goal | 40.9 | 12.65 | 45.26 | 1.2 | 41.94 | 9.81 | 47.19 | 1.06 | 49.33 | 8.55 | 40.7 | 1.42 |
| | Persona Consistency | 33.2 | 11.62 | 53.95 | 1.24 | 36.23 | 7.06 | 55.52 | 1.18 | 39.64 | 4.41 | 54.75 | 1.2 |
| | Coherence | 59.4 | 21.08 | 18.12 | 1.4 | 56.49 | 20.91 | 21.33 | 1.27 | 62.62 | 17.19 | 19.17 | 1.01 |
| | Plot Conclusion | 49.14 | 22.97 | 26.65 | 1.24 | 50.59 | 19.16 | 28.8 | 1.45 | 51.93 | 16.89 | 30.05 | 1.13 |
| **Knowledge Base** | Depth of Analysis | 58.67 | 21.75 | 18.46 | 1.12 | 58.18 | 19.59 | 21.05 | 1.18 | 60.89 | 17.1 | 20.77 | 1.24 |
| | Relevance To Goal | 41.27 | 12.95 | 44.35 | 1.43 | 41.2 | 12.68 | 44.65 | 1.47 | 48.21 | 7.41 | 43.1 | 1.28 |
| | Persona Consistency | 35.8 | 8.23 | 54.86 | 1.12 | 36.95 | 8.5 | 53.05 | 1.5 | 40.12 | 8.23 | 50.19 | 1.46 |
| | Coherence | 56.41 | 21.15 | 21.0 | 1.43 | 55.75 | 21.44 | 21.32 | 1.49 | 62.33 | 17.6 | 18.98 | 1.09 |
| | Plot Conclusion | 49.18 | 20.56 | 29.09 | 1.16 | 52.13 | 18.89 | 27.9 | 1.08 | 52.22 | 15.59 | 30.81 | 1.38 |
| **Huge Table Analysis** | Depth of Analysis | 56.09 | 21.64 | 21.2 | 1.07 | 56.75 | 21.27 | 20.79 | 1.19 | 61.54 | 17.31 | 20.11 | 1.04 |
| | Relevance To Goal | 37.01 | 12.52 | 49.21 | 1.26 | 42.99 | 10.92 | 44.8 | 1.29 | 46.35 | 9.53 | 43.07 | 1.04 |
| | Persona Consistency | 32.48 | 12.32 | 53.83 | 1.37 | 35.43 | 7.62 | 55.46 | 1.49 | 40.33 | 5.6 | 52.64 | 1.43 |
| | Coherence | 57.36 | 21.39 | 19.93 | 1.32 | 56.7 | 20.63 | 21.38 | 1.29 | 62.19 | 16.96 | 19.55 | 1.3 |
| | Plot Conclusion | 48.77 | 22.46 | 27.32 | 1.45 | 50.71 | 19.76 | 28.09 | 1.44 | 54.2 | 15.1 | 29.66 | 1.04 |
| **Topic Modeling** | Depth of Analysis | 56.91 | 22.43 | 19.28 | 1.38 | 58.78 | 19.21 | 20.99 | 1.03 | 62.64 | 15.57 | 20.31 | 1.47 |
| | Relevance To Goal | 39.71 | 13.87 | 45.24 | 1.18 | 44.09 | 11.4 | 43.41 | 1.19 | 49.15 | 9.92 | 39.57 | 1.36 |
| | Persona Consistency | 31.72 | 8.71 | 58.22 | 1.35 | 37.36 | 7.35 | 53.83 | 1.46 | 41.0 | 4.71 | 52.83 | 1.46 |
| | Coherence | 57.48 | 21.32 | 20.14 | 1.06 | 57.0 | 21.61 | 20.07 | 1.32 | 60.6 | 17.71 | 20.36 | 1.33 |
| | Plot Conclusion | 47.9 | 20.95 | 29.92 | 1.23 | 51.54 | 21.09 | 25.98 | 1.39 | 54.15 | 15.22 | 29.4 | 1.23 |
| **Market Analysis** | Depth of Analysis | 56.22 | 22.87 | 19.91 | 1.0 | 56.93 | 19.18 | 22.57 | 1.32 | 62.91 | 15.06 | 20.55 | 1.49 |
| | Relevance To Goal | 39.37 | 12.66 | 46.5 | 1.47 | 43.21 | 8.92 | 46.55 | 1.32 | 47.27 | 7.41 | 44.02 | 1.3 |
| | Persona Consistency | 34.29 | 9.39 | 54.84 | 1.47 | 34.7 | 7.87 | 56.34 | 1.09 | 39.02 | 8.03 | 51.71 | 1.24 |
| | Coherence | 56.42 | 24.49 | 18.05 | 1.04 | 58.33 | 20.55 | 19.83 | 1.29 | 61.53 | 15.44 | 21.75 | 1.28 |
| | Plot Conclusion | 49.56 | 23.5 | 25.67 | 1.27 | 50.03 | 20.28 | 28.19 | 1.5 | 52.21 | 17.34 | 29.37 | 1.07 |
| **Data Imputation** | Depth of Analysis | 56.91 | 24.12 | 17.84 | 1.13 | 58.8 | 21.08 | 19.07 | 1.06 | 62.37 | 16.74 | 19.75 | 1.14 |
| | Relevance To Goal | 37.8 | 12.78 | 47.99 | 1.43 | 42.31 | 9.83 | 46.67 | 1.19 | 48.43 | 7.9 | 42.56 | 1.11 |
| | Persona Consistency | 33.6 | 7.34 | 57.95 | 1.11 | 33.67 | 9.2 | 55.67 | 1.46 | 42.26 | 4.81 | 51.54 | 1.39 |
| | Coherence | 56.13 | 24.08 | 18.47 | 1.32 | 58.41 | 19.56 | 20.82 | 1.21 | 62.11 | 14.52 | 21.99 | 1.37 |
| | Plot Conclusion | 47.99 | 22.49 | 28.18 | 1.34 | 49.03 | 19.78 | 29.68 | 1.5 | 53.06 | 16.24 | 29.66 | 1.04 |
| **Multi-table Search** | Depth of Analysis | 59.17 | 21.29 | 18.43 | 1.12 | 59.43 | 20.48 | 19.07 | 1.02 | 62.72 | 15.49 | 20.42 | 1.37 |
| | Relevance To Goal | 40.42 | 11.46 | 46.85 | 1.27 | 41.85 | 11.19 | 45.95 | 1.01 | 46.55 | 9.01 | 43.0 | 1.44 |
| | Persona Consistency | 32.07 | 11.67 | 55.26 | 1.0 | 37.89 | 7.17 | 53.85 | 1.08 | 40.27 | 6.4 | 52.0 | 1.33 |
| | Coherence | 55.53 | 22.31 | 20.98 | 1.18 | 58.0 | 20.78 | 19.75 | 1.47 | 61.51 | 14.77 | 22.62 | 1.1 |
| | Plot Conclusion | 52.04 | 21.73 | 25.15 | 1.08 | 51.74 | 20.18 | 26.59 | 1.49 | 54.3 | 15.89 | 28.43 | 1.38 |

