# OpenReview forum: "AgentAda: Skill-Adaptive Data Analytics for Tailored Insight Discovery"
_ICLR.cc/2026/Conference — ICLR 2026 Conference Withdrawn Submission_

### Official Review · Reviewer_K82t · 2025-10-25

**Soundness:** 2
**Presentation:** 2
**Contribution:** 1
**Rating:** 2
**Confidence:** 4

**Summary:**

This paper introduces AgentAda, a skill-adaptive LLM-based data analytics agent that retrieves analytical “skills” from a curated library (e.g., clustering, regression, topic modeling) to produce goal-aligned insights. The pipeline includes four stages: question generation, skill retrieval via hybrid RAG, code generation, and insight extraction. The authors also present two supporting resources: KaggleBench: a large benchmark derived from Kaggle notebooks for evaluating analytic reasoning; SCORER: a “prompt-optimized LLM-as-a-judge” method aligning automated scoring with human preferences. Empirically, AgentAda reportedly outperforms baseline analytics agents (e.g., Poirot, InfiAgent, MetaGPT, PandasAI) in human and LLM-judge evaluations across different rubrics.

**Strengths:**

- AgentAda integrates question generation, skill retrieval, code execution, and insight summarization into a unified pipeline.
- A new benchmark is proposed. KaggleBench forms large, realistic dataset for analytic reasoning, potentially reusable by the community.
- Strong empirical section. The paper includes comparisons with multiple baselines, qualitative examples, and factuality analysis.

**Weaknesses:**

- The notion of retrieving or matching analytical “skills” closely parallels well-established RAG-augmented frameworks (e.g., ReAct, Data Interpreter, InfiAgent). The incremental improvement (structured skill library + two-stage question generation) is not convincingly shown to yield qualitatively new behavior.
- While the system’s pipeline is technically sound, the notion of “insight” remains an intuitive amalgam of answer summaries rather than a theoretically or empirically grounded construct. The authors could strengthen the paper by explicitly connecting their definition of insight to prior models of analytical sense-making and pattern discovery in the data management and visualization literature. Just to list a few:
  - Towards a Unified Representation of Insight in Human-in-the-Loop Analytics: A User Study. HILDA 2018
  - What exactly is an insight? a literature review. IEEE VIS 2023
  - Characterizing the quality of insight by interactions: A case study. IEEE TVCG
- The SCORER module, while presented as a novel “prompt-optimized LLM-as-a-judge,” feels incremental relative to prior evaluation frameworks such as Prometheus, JudgeLM, and InstructScore. Conceptually, it replaces supervised fine-tuning with prompt-optimization (via TextGrad) but otherwise retains the same structure and purpose, i.e., aligning LLM judgments with human preferences. The paper does not clearly articulate the trade-offs of this substitution: whether it meaningfully reduces cost, maintains calibration quality, or generalizes across domains. Without comparative evidence or quantitative analysis, SCORER appears to be a lightweight engineering variant of existing fine-tuning-based evaluators rather than a substantial contribution.
- There's no mention about the IRB approval to the human evaluation.

**Questions:**

1. How do the authors formally define an “insight” in AgentAda, and how does this definition connect to prior models of analytical sense-making and pattern discovery in the data management and visualization communities?
2. What empirical or theoretical justification supports the choice of TextGrad-based prompt optimization over supervised fine-tuning?
3. Also, please clarify whether the human evaluation study received IRB or equivalent ethics-board approval.

**Details Of Ethics Concerns:**

There's no mention about the IRB approval to the human evaluation.

---

### Official Review · Reviewer_mViZ · 2025-10-28

**Soundness:** 3
**Presentation:** 3
**Contribution:** 3
**Rating:** 6
**Confidence:** 2

**Summary:**

This paper (i) introduces AGENTADA, a skill-informed data analytics agent that dynamically selects skills from a curated library and generates executable code to produce goal-aligned insights for advanced, diverse tasks; (ii) It releases KAGGLEBENCH, a 700-example benchmark spanning 49 domains and 28 task types, reflecting the complexity and diversity of real-world analysis; (iii) It proposes SCORER, a prompt-optimized LLM-as-a-judge framework that, via expert-guided supervision, aligns automatic evaluation with human judgments of analytical quality; (iv) It provides comprehensive evaluations showing AGENTADA surpasses existing agents in analytical depth and in alignment with task goals and user personas.

**Strengths:**

1. The proposed skill set matching technique is quite effective and suitable for the nature fo data analysis task.
2. The proposed AgentAda surpasses existing agents in data analysis task.
3. The proposed KaggleBench could be useful for the community.

**Weaknesses:**

1. AgentAda is a staged workflow which consisting steps such as Question Generation, Skill-matching, Code Generation, Answer Generation, Category prediction and Insight Generation. However, the impact of certain steps are not ablated in this paper, e.g., question generation category prediction.
2. Lack of experiments with large reasoning models (LRM). The data analysis task might benefit from deep reasoning with LRM. However, the LLMs adopted only include general instruction-following LLMs. Can the authors include LRM in certain staegs (e.g., code generation) to see its impact?
3. Lack of evaluations and ablations on SCORER. To evaluate the proposed SCORER, the authors directly report win/tie/lose scores from SCORER and human evaluation. However, this alone does not indicate how closely they are aligned. The authors should also report agreement scores [1] for these two evaluators. In addition, the authors should compare the performance of the judge models when it is not optimized with TextGrad to see the necessity of optimization. By the way, could the authors provide some details on TextGrad?


References:
[1] Lianmin Zheng, Wei-Lin Chiang, Ying Sheng, Siyuan Zhuang, Zhanghao Wu, Yonghao Zhuang, Zi Lin, Zhuohan Li, Dacheng Li, Eric Xing, et al. Judging LLM-as-a-judge with mt-bench and chatbot arena. In NeurIPS, 2023.

**Questions:**

1. In the caption of Table 4 (lines 397-398), WO denotes winning by the baseline agent while, WO refers to  W/O skill in Table 3. What is the difference between WO and baseline agent?
2. How many LLMs/MLLMs are used in the AgentAda framework? In lines 250-251, GPT-4o is used to read the generated plots, are the LLMs that perform other steps also GPT-4o?
3. What LLMs are used for the judge models? Are they open-sourced or API models?

---

### Official Review · Reviewer_n55S · 2025-10-30

**Soundness:** 1
**Presentation:** 3
**Contribution:** 1
**Rating:** 2
**Confidence:** 4

**Summary:**

This paper introduces AgentAda, an agentic data analysis framework, as well as a data analysis benchmark (KaggleBench) and an LLM-as-a-judge scoring mechanism (SCORER).

**Strengths:**

The paper is relatively clear, the skill matcher component of AgentAda is an effective intervention, and releasing KaggleBench/SCORER openly will be helpful to the community.

**Weaknesses:**

My main issue with this work is the lack of benchmarking outside of KaggleBench/SCORER, which are contributed by the same work. I would strongly advocate for additional experiments with existing benchmarks, even though the authors make the case that they are less complete than KaggleBench. Additionally, the only non-SCORER-based eval is human evaluation on the "w/ skill" vs "w/o skill" variants of AgentAda, but this doesn't include any baseline methods that were evaluated via SCORER (e.g., Pandas AI, Poirot).

**Questions:**

- How did the authors verify that there is no leakage between the skill library and KaggleBench?
- Table 4 is missing “answers questions adequately” row?
- What prompt is used for gpt-4o baseline?

---

### Official Review · Reviewer_PqZp · 2025-11-01

**Soundness:** 3
**Presentation:** 3
**Contribution:** 2
**Rating:** 4
**Confidence:** 4

**Summary:**

This paper introduces AgentAda, an LLM-powered data analytics agent that dynamically retrieves and applies specialized analytical skills from a curated library of 74 methods to generate deeper, goal-aligned insights. The authors also contribute KaggleBench and SCORER benchmarks to comprehensively assess the effectiveness of the proposed method.

**Strengths:**

1. The paper presents an interesting and practical approach to constructing a reusable skill library by extracting 74 analytical workflows from Kaggle notebooks and converting them into structured text descriptions that guide LLMs to perform advanced analytics beyond their native capabilities.
2. Comprehensive experiments show the effectiveness of the proposed method.
3. The proposed KaggleBench and SCORER provide valuable research infrastructure for subsequent research.

**Weaknesses:**

1. The novelty and contribution of the work need clarification, as there already exist similar benchmarks for data analytics tasks and skill-based agent frameworks that leverage retrieval-augmented generation.
2. The construction and validation of the skill library lack sufficient justification. The paper does not provide clear criteria or analysis for determining the appropriate granularity of skills (e.g., why 74 skills, why these specific decompositions), nor does it validate whether the extracted skills are comprehensive, non-redundant, and optimally defined. The effectiveness of individual skills and the rationale for skill selection from Kaggle notebooks may not have been thoroughly examined.
3. The scalability and maintenance of the skill library present practical concerns. As data analytics methods evolve rapidly with new algorithms and techniques, the paper does not adequately address how the skill library would be updated, extended, or quality-controlled over time. Additionally, the heavy reliance on GPT-4o throughout the pipeline raises questions about cost-effectiveness and whether the approach would remain practical for real-world deployment at scale.

**Questions:**

See weaknesses.

---

### Note · Authors · 2025-11-17

**Comment:**

The authors would like to thank the Area Chair and the reviewers for their time and constructive feedback. After carefully reviewing the comments, we have decided to withdraw this submission to incorporate the suggested improvements and conduct additional experiments. We believe these changes will significantly strengthen the paper for a future submission. We appreciate the effort put into the review process.

**Withdrawal Confirmation:**

I have read and agree with the venue's withdrawal policy on behalf of myself and my co-authors.